# Fluorene-9-bisphenol is anti-oestrogenic and may cause adverse pregnancy outcomes in mice

Zhaobin Zhang[1], Ying Hu[1,2], Jilong Guo[1], Tong Yu[1], Libei Sun[1], Xuan Xiao[1], Desheng Zhu[3], Tsuyoshi Nakanishi[4], Youhei Hiromori[4,5], Junyu Li[1], Xiaolin Fan[1], Yi Wan[1], Siyu Cheng[1], Jun Li[3], Xuan Guo[1] & Jianying Hu[1]

Bisphenol A (BPA) is used in the production of plastic but has oestrogenic activity. Therefore, BPA substitutes, such as fluorene-9-bisphenol (BHPF), have been introduced for the production of so-called 'BPA-free' plastics. Here we show that BHPF is released from commercial 'BPA-free' plastic bottles into drinking water and has anti-oestrogenic effects in mice. We demonstrate that BHPF has anti-oestrogenic activity *in vitro* and, in an uterotrophic assay in mice, induces low uterine weight, atrophic endometria and causes adverse pregnancy outcomes, even at doses lower than those of BPA for which no observed adverse effect have been reported. Female mice given water containing BHPF released from plastic bottles, have detectable levels of BHPF in serum, low uterine weights and show decreased expressions of oestrogen-responsive genes. We also detect BHPF in the plasma of 7/100 individuals, who regularly drink water from plastic bottles. Our data suggest that BPA substitutes should be tested for anti-oestrogenic activity and call for further study of the toxicological effects of BHPF on human health.

[1] College of Urban and Environmental Sciences, MOE Laboratory for Earth Surface Process, Peking University, Beijing 100871, China. [2] College of Chemistry and Environmental Engineering, Shenzhen University, Shenzhen, Guangdong 518060, China. [3] Laboratory Animal Centre, Peking University, Beijing 100871, China. [4] Laboratory of Hygienic Chemistry and Molecular Toxicology, Gifu Pharmaceutical University, 1-25-4 Daigaku-nishi, Gifu, Gifu 501-1196, Japan. [5] Faculty of Pharmaceutical Sciences, Suzuka University of Medical Science, 3500-3, Minamitamagaki, Suzuka, Mie 513-8670, Japan. Correspondence and requests for materials should be addressed to D.Z. (email: deshengz@pku.edu.cn) or to J.H. (email: hujy@urban.pku.edu.cn).

The chemical bisphenol A (BPA) is produced in large quantities for the production of polycarbonate (PC) and epoxy resin plastics, which are used various daily life products[1–3]. In recent years, BPA was shown to have oestrogenic activity, linking BPA to endocrine diseases and to an increased incidence of endocrine-related cancers[3–9]. Because materials synthesized with BPA were widely used in packing materials for food and beverages, and BPA can be released into food from such containers[1–3], many countries have restricted or banned the use of BPA in materials or containers that come in contact with food, especially baby bottles for milk or water. This ban has led to the introduction of BPA substitutes by the plastics industry.

One such BPA substitute is, fluorene-9-bisphenol (CAS NO. 3236-71-3), also known as 9,9-bis(4-hydroxyphenyl)-fluorene (BHPF). BHPF is used in the synthesis of polyester polymers such as PC, epoxy resins, polyurethanes, polyesters, polyarylates and polyethers[10–13]. In recent years, materials containing BHPF have been used to produce a variety of products, including protective coatings, semiconductor encapsulations, composite matrices, moulded products, insulation materials, solder-resistant materials, epoxy floor coatings and structural adhesives, in the electronics, aerospace, automobile and other industries. However, whether BHPF is also used in materials or containers that come into contact with food—including milk bottles, children's bottles and sippy cups—and whether humans are exposed to BHPF remains unclear.

Over the past decades, studies[2,3,14–16] have focused on the oestrogenicity of BPA substitutes and related compounds, attributing exposure to such compounds with endocrine-related diseases, such as the loss of gender characteristics, precocious puberty, low semen quality and obesity. However, the anti-oestrogenicity of environmental pollutants, as well as the potential adverse effects thereof, have rarely been studied. Because oestrogens play a critical role in maintaining the development of female reproductive organs and the course of pregnancy, anti-oestrogenic effects may be physiologically important. Anti-oestrogenic drugs, such as tamoxifen (TAM) and U39411, have been reported to prevent pregnancy, reduce litter size and weight, and induce embryonic absorption in animals. According to a report by WHO-UNEP entitled 'State of the Science of Endocrine Disrupting Chemicals—2012,' (ref. 17) significant knowledge gaps exist on the association between exposure to endocrine-disrupting chemicals and adverse pregnancy outcomes, even though the incidence of these outcomes, such as spontaneous abortion, stillbirth, low birth weight and fetal death, has increased in many countries over the last decades. Therefore, assessing the anti-oestrogenicity of environmental pollutants and evaluating their effects on female development and reproduction is important.

In the present study, we report the presence of BHPF in plastic bottles, measure BHPF concentrations in water released from commercial plastic bottles and in serum samples of human volunteers, determine the anti-oestrogenic activity of BHPF by in vitro, in vivo and in in silico assays, and evaluate the developmental and reproductive toxicities of BHPF using mice following subchronic exposure. We believe that this study will extend the knowledge of environmental anti-oestrogens, and raise awareness of anti-oestrogenic BPA substitutes and the possibility of their causing adverse effects on human health.

## Results

### Determination of BHPF released from plastic bottles. 

BHPF was identified in a fraction isolated from the methanol leachates of plastic drink bottles labelled 'BPA-free' using $^1H$ and $^{13}C$ nuclear magnetic resonance (NMR) analysis (Fig. 1). The NMR results are consistent with those for BHPF reported previously[11]. After synthesis of a deuterated standard of BHPF (BHPF-$d_7$) (Methods, Supplementary Fig. 1a,b), a reliable derivatization method for the sensitive quantification of BHPF in water by gas chromatography-mass spectrometry (GC–MS) was developed (Methods, Supplementary Fig. 1c). We then studied the release of BHPF from 52 commercial plastic bottles into drinking water. The concentrations of BHPF in water samples that had been contained in bottles of different materials are summarized in Supplementary Table 1. The release of BHPF was detectable in 61.11% (11/18) of the water samples from PC bottles, with a maximal concentration of 81.47 $ng\,l^{-1}$ from a PC water bottle intended for adults. Because PC bottles for babies' milk are now prohibited in China, we purchased three PC bottles from non-Chinese suppliers, and found that BHPF was detectable in the water samples from these bottles. In addition, BHPF was also detected in water from two bottles made of Eastman Tritan copolyester, a BPA-free material.

### BHPF in serum samples from college student volunteers. 

Because BHPF is widely used in various products in daily life, and human exposure occurs through drinking water, we further determined the BHPF levels in serum samples from 100 college student volunteers (50 male and 50 female) who habitually used plastic bottles for drinking water. BHPF was detectable in seven volunteers (four male and three female), with a mean serum BHPF level of $0.34 \pm 0.21\,ng\,ml^{-1}$ and a maximal level of $0.7\,ng\,ml^{-1}$.

### Anti-oestrogenic activity of BHPF in vitro. 

The oestrogenic and anti-oestrogenic activities of BHPF were studied by a yeast two-hybrid assay and dual-luciferase reporter assays. The yeast two-hybrid assay comprised the ligand-binding domain of human oestrogen receptor α and a co-activator. The dual-luciferase reporter assay comprised a full-length human oestrogen receptor α or β and oestrogen response elements (EREs). As shown in Fig. 2a, 17β-estradiol (E₂) increased the β-galactosidase activity in the yeast, with a half-maximal effect concentration of $1.28 \times 10^{-9}\,M$, but BHPF did not show oestrogenic activity; however, when BHPF coexisted with $1 \times 10^{-9}\,M$ E₂, BHPF showed strong anti-oestrogenic activity in a concentration-dependent manner, and the half-inhibition concentration (IC₅₀) of BHPF was $5.44 \times 10^{-7}\,M$, which was close to that $(4.50 \times 10^{-7}\,M)$ of 4-hydroxytamoxifen (OHT) (Fig. 2b). Similarly, BHPF showed no oestrogenic activity but strong anti-oestrogenic activities in the dual-luciferase reporter assays (Fig. 2c–f). The IC₅₀ values of BHPF were $1.09 \times 10^{-7}$ and $7.53 \times 10^{-8}\,M$ for oestrogen receptor α and oestrogen receptor β, respectively, when BHPF coexisted with $1 \times 10^{-9}\,M$ E₂. A WST-1 cell proliferation assay using human choriocarcinoma JEG-3 cells was performed, and no obvious cytotoxicity of BHPF was observed from $1.0 \times 10^{-6}\,M$ to lower concentrations (Supplementary Fig. 2). The IC₅₀ values of OHT were $8.61 \times 10^{-9}$ and $5.19 \times 10^{-9}\,M$ for oestrogen receptor α and oestrogen receptor β, respectively, which were 12.66 and 14.51 times lower than those of BHTF in the dual-luciferase DLR assays.

### Docking of BHPF into active sites of oestrogen receptor α. 

Molecular docking was performed to investigate the structural basis of the observed anti-oestrogenic activity of BHPF, using Scigress (Ultra Version 3.0.0, Fujitsu). We found that BHPF could not be accommodated by the ligand pockets of oestrogen receptor α (PDB IDs 1ERE, 3UU7, 3UUA and 3UUC), but BHPF could be well fitted into the antagonist pocket of oestrogen receptor α

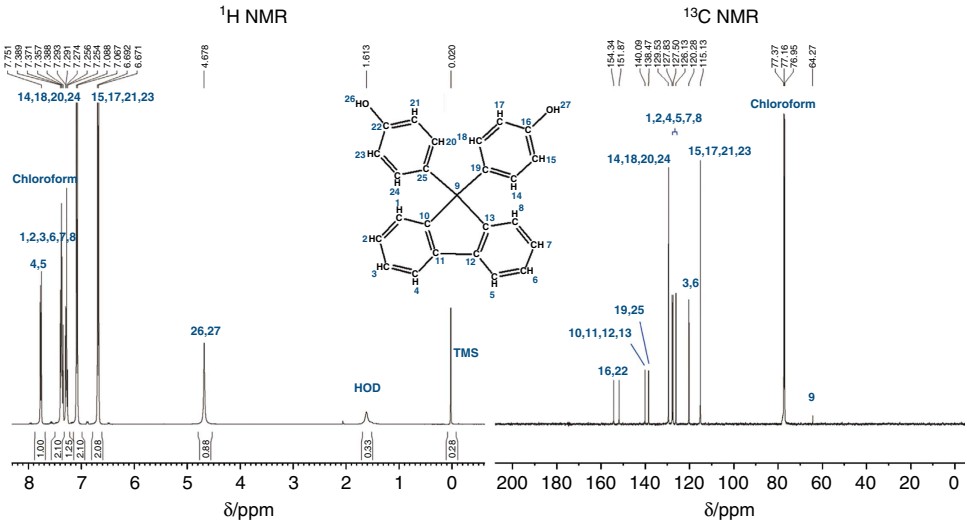

**Figure 1 | $^1$H and $^{13}$C NMR spectra for the identification of BHPF leached from a BPA-free plastic bottle.** $^1$H NMR (400 MHz, chloroform-d, TMS, p.p.m.). δ; 7.75 (d, 2H, H4, H5), 7.25–7.39 (m, 6H, H1, H2, H3, H6, H7, H8), 7.27 (CDCl3), 7.06–7.09 (d, 4H, H14, H18, H20, H24), 6.67–6.70 (d, 4H, H15, H17, H21, H23), 4.67 (s, 2H, H26, H27), 1.61 (HOD), 0.02 (TMS); $^{13}$C NMR (600 MHz, methanol-d$_4$, p.p.m.). δ; 151.81–155.32 (2C: C16, C22), 139.50 (4C, C10, C11, C12, C13), 136.60 (2C: C19, C25), 128.44 (4C, C14, C18, C20, C24), 125.41–126.69 (6C, C1, C2, C4, C5, C7, C8), 119.19 (2C, C3, C6), 114.00 (4C, C15, C17, C21, C23), 64.27 (1C, C9), 47.64–46.79 (1C, carbon atom signal of MeOD).

(PDB ID 3ERT). The interaction between BHPF and the antagonist pocket of oestrogen receptor α is shown in Fig. 2g. The important features of the interaction are (a) the formation of hydrogen bonds between a phenolic hydroxyl group of BHPF and the carboxylate of Glu-353, the guanidinium group of Arg-394 and a water molecule (H$_2$O 2) of the pocket—and (b) the van der Waals forces between the phenyl rings of BHPF and the core hydrophobic moiety provided by Leu-328, Met-342, Met-343, Leu-346, Leu-349, Leu-384, Leu-387, Met-388, Leu-391, Phe-404, Leu-410, Val-418, Met-421, Ile-424, Met-517 and Leu-525. As shown in Fig. 2h, the optimal position of BHPF could be well accommodated by the antagonist pocket of oestrogen receptor α, with one of the phenol groups of BHPF located at the site for the phenol group of OHT (the original ligand of the structure (3ERT)), the other phenol group of BHPF located at the site for the phenyl ring on the side chain of OHT, and the fluorene group in the space for the 1-phenylpropylidene of OHT.

**Anti-oestrogenic activity of BHPF in vivo.** The uterotrophic bioassay in rodents is a short-term in vivo screening test for oestrogenic substances, which has occasionally been used to determine anti-oestrogenic compounds by combination with exogenous oestrogen. As shown in Fig. 3a, BHPF showed neither uterotrophic nor anti-uterotrophic effects when singly administered to mice for 3 days beginning on postnatal day (PND) 21; however, when combined with E$_2$, it significantly ($P < 0.01$, Fisher's least significant difference (LSD) test) inhibited the uterotrophic effect of E$_2$ in the 250 and 500 mg kg$^{-1}$ d$^{-1}$ groups (Fig. 3b). Considering that endogenous oestrogens are increasingly produced when mice reach puberty, an improved anti-uterotrophic assay for evaluating the anti-oestrogenicity of a chemical using female CD-1 mice ($n = 10$) at a beginning age (PND 24) close to puberty were developed by daily oral gavage of fulvestrant (FULV) for 10 days. As shown in Fig. 3c, after 10 days of treatment, the relative uterine weights were decreased in a dose-dependent manner in the mice given FULV, indicating that this anti-uterotrophic assay should be suitable for evaluating the anti-oestrogenicity of chemicals. Using the improved assay, the relative uterine weights were found to be significantly decreased ($P < 0.05$, Fisher's LSD test) to ~63.20, 59.59 and 52.80% that of

the control in mice given BHPF at doses of 27.8, 83.3 and 250 mg kg$^{-1}$ body weight (bw) per day, respectively (Fig. 3d). The anti-uterotrophic activity of BHPF at a dose of 27.8 mg kg$^{-1}$ bw d$^{-1}$ was close to that of FULV at 2 mg kg$^{-1}$ bw d$^{-1}$.

**Opposite patterns of gene regulation by BHPF and E$_2$.** Analysis of the global gene expression profiles in the livers, ovaries and uteri of mice treated with 50 mg kg$^{-1}$ bw d$^{-1}$ of BHPF for 3 days compared with those of the control mice was performed using an Agilent mouse gene expression microarray (Agilent Technologies). In the livers, the expressions of 2,963 of the 55,821 genes and expressed-sequence tags (ESTs) evaluated was modified by at least two-fold in mice after BHPF treatment (up-regulated in 77.7% and down-regulated in 22.3%). GO analysis showed that 1,363 genes were enriched in 207 biological process categories, such as defence responses, responses to stimuli, responses to other organisms, multi-organism processes, immune system processes, responses to biotic stimuli and inflammatory responses. Pathway enrichment analyses showed that 302 altered genes in the liver were enriched in 29 pathways, such as natural killer cell-mediated cytotoxicity, hematopoietic cell lineage and autoimmune thyroid disease. In the ovaries, the expressions of 948 genes and ESTs was up-regulated (34.5%) or down-regulated (65.5%) by at least two-fold after BHPF treatment. Among them, 488 genes were enriched in 45 biological process categories, including immune system processes, responses to stimuli, responses to E$_2$ stimuli, responses to chemical stimuli, and immune responses. The genes associated with 'response to E$_2$ stimuli' were sprr2a, sprr2b, sprr2f and sprr2g, which were down-regulated by BHPF. Pathway analyses showed that 98 genes were enriched in 14 pathways, including natural killer cell-mediated cytotoxicity, viral myocarditis, autoimmune thyroid disease, and the B-cell receptor-signaling pathway. In the uteri, the expressions of 1324 genes or ESTs was up-regulated (56.6%) or down-regulated (43.4%) by at least two-fold after BHPF treatment. Of them, 534 genes were enriched in 200 biological processes, including responses to stimuli, responses to chemical stimuli, responses to E$_2$ stimuli, oxidation-reduction processes, cholesterol homoeostasis and efflux, regulation of cholesterol

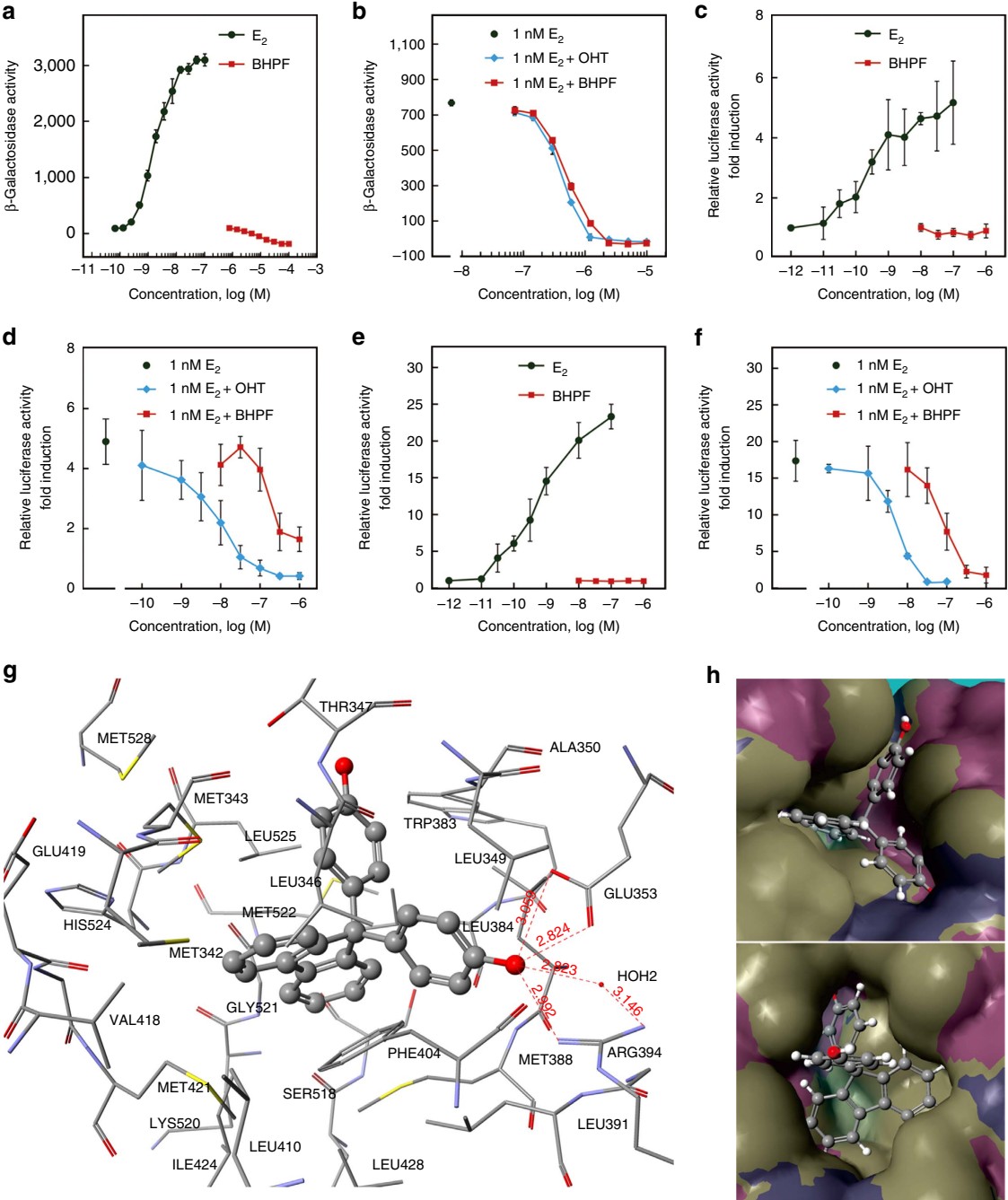

**Figure 2 | Anti-oestrogenic activity of BHPF shown by *in vitro* and *in silico* assays.** (**a**) No oestrogenic activities of BHPF were observed by yeast two-hybrid assay. (**b**) The anti-oestrogenic activity of BHPF in comparison with that of OHT was shown by yeast two-hybrid assay. (**c**) No oestrogenic activities of BHPF were observed by dual-luciferase reporter assay for oestrogen receptor α. (**d**) The anti-oestrogenic activity of BHPF in comparison with that of OHT was shown by dual-luciferase reporter assay for oestrogen receptor α. (**e**) No oestrogenic activities of BHPF were observed by dual-luciferase reporter assay for oestrogen receptor β. (**f**) The anti-oestrogenic activity of BHPF in comparison with that of OHT was shown by dual-luciferase reporter assay for oestrogen receptor β. (**g**) Interaction between BHPF and the active site in human oestrogen receptor α structure (3ERT) for antagonist; dotted lines indicate hydrogen bonds between the chemical and amino acid residues. (**h**) Simulated binding pose of BHPF in the antagonist pocket of human oestrogen receptor α structure (3ERT) viewed from different angles. The yeast two-hybrid assays were performed in triplicate; the dual-luciferase reporter assays were performed in quadruplicate. Error bars indicate the s.d.'s from the average.

absorption, acute-phase responses and steroid metabolic processes. Pathway analyses showed that 228 genes were enriched in 27 pathways, including complement and coagulation cascades, drug metabolism, retinol metabolism, metabolism of xenobiotics by cytochrome P450, steroid hormone biosynthesis, and androgen and oestrogen metabolism. The expression profile of oestrogen-responsive genes in the uteri is of great concern, so we

compared our gene expression data with an expression profile of oestrogen-responsive genes in the uteri of mice administered $E_2$ from previously published data[18]. As shown in Fig. 4a, most of the oestrogen-responsive genes exhibited the opposite expression pattern in the uteri of mice treated with BHPF to that induced by $E_2$. Of particular note is the down-regulation of small proline-rich proteins (SPRR), including *Sprr2a2*, *Sprr2b*, *Sprr2d*, *Sprr2f* and

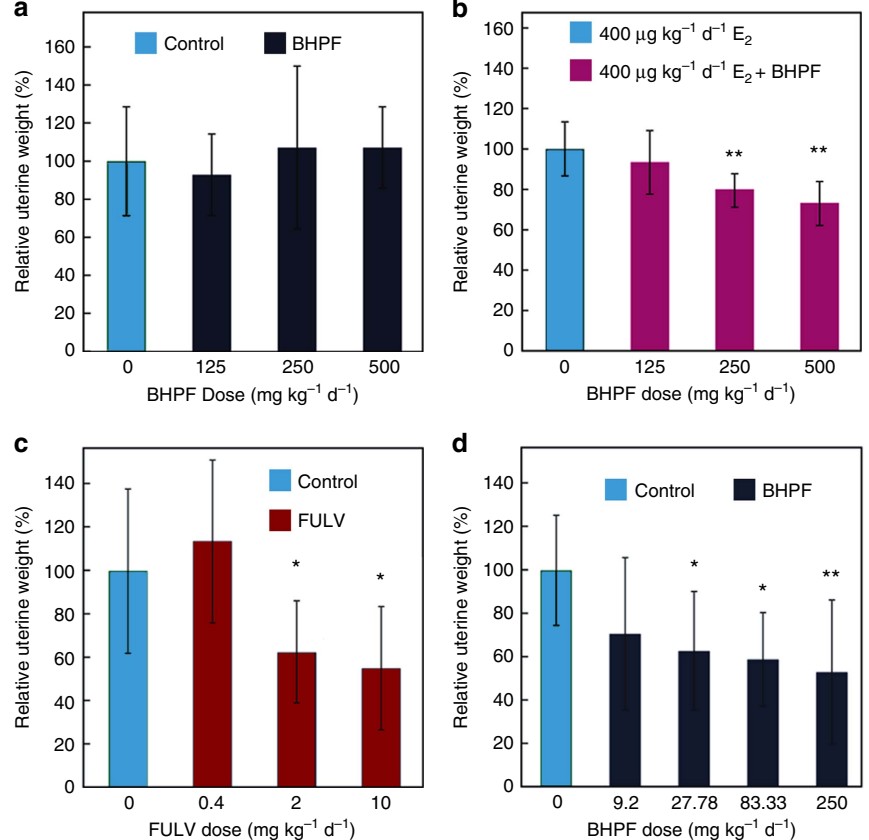

**Figure 3 | Uterotrophic/anti-uterotrophic effects of BHPF on CD-1 mice.** (**a**) Relative uterine weights of CD-1 mice given BHPF for 3 days beginning on PND 21. (**b**) Relative uterine weights of CD-1 mice given BHPF in combination with 400 μg kg$^{-1}$ bw d$^{-1}$ of $E_2$ for 3 days beginning on PND 21; a group receiving 400 μg kg bw d$^{-1}$ of $E_2$ was used as a control. (**c**) Relative uterine weights of CD-1 mice given FULV daily for 10 days beginning on PND 24. (**d**) Relative uterine weights of CD-1 mice given BHPF daily for 10 days beginning on PND 24. Ten mice per group were used for the assays. Error bars indicate the s.d.'s from the average. $*P < 0.05$ compared with the corresponding control. $**P < 0.01$ compared with the corresponding control. Significance was tested by Fisher's LSD test.

*Sprr2g*, in the uteri. In addition, we found that many genes involved in the biotransformation and oxidation-reduction processes were up-regulated in the uterus but not in the liver (Fig. 4b). The expressions of 14 genes were quantified to validate the microarray by real-time quantitative reverse-transcription polymerase chain reaction (Q-RT-PCR), and it was found that most genes showed similar results to those of the microarrays (Supplementary Table 2). We selected *sprr2a* and *sprr2b* as biomarker genes and studied their expressions in the uteri of the mice given BHPF at doses of 0, 9.26, 27.8, 83.3 and 250 mg kg$^{-1}$ bw d$^{-1}$ for 10 days. As shown in Fig. 4c, the expressions of *sprr2a* and *sprr2b* were significantly decreased in a dose-dependent manner; both genes were significantly down-regulated by 9.26 mg kg$^{-1}$ bw d$^{-1}$ BHPF and higher doses ($P < 0.05$, Fisher's LSD test).

**Subchronic and reproductive toxicity of BHPF in mice.** To identify the subchronic and reproductive toxicity associated with the anti-oestrogenicity of BHPF, mice were given doses of 0.4, 2, 10 and 50 mg kg$^{-1}$ bw 3-d$^{-1}$ BHPF or 1.2 mg kg$^{-1}$ bw 3-d$^{-1}$ TAM by oral gavage once every 3 days from PND 24. The body weight evolution of the mice during the experiments is shown in Fig. 5a,b. Before pairing, no significant differences in weight gain were found in relation to the administration of these substances, except in the male receiving 0.4 mg kg$^{-1}$ 3-d$^{-1}$ BHPF, which showed higher weight gain than the control from PND 30. After mating, pregnancy-related weight gain was significantly decreased

in all of the BHPF-treated groups, but no pregnancy-related weight gain was observed in the TAM-treated group because all of the females were non-pregnant. The data obtained from the subchronic and reproductive toxicity tests are summarized in Table 1 and Supplementary Tables 5–7. After 36-day exposure, 10 females of each group were weighed and killed on PND 60. The relative uterine weights were found to be decreased in a dose-dependent manner in the mice that had been given BHPF (Table 1). The relative uterine weight in the 2, 10 and 50 mg kg$^{-1}$ 3-d$^{-1}$ BHPF groups and the 1.2 mg kg$^{-1}$ 3-d$^{-1}$ TAM group were significantly ($P < 0.05$, Fisher's LSD test) lower than that of the control. Histopathological examination found serious atrophic endometria in the uteri of mice administered BHPF and TAM. Figure 5c shows microscopic images of uterine sections. The endometrial stromal cells and columnar epithelial cells were observed to be atrophic in each of the mice treated with BHPF, but in the TAM-treated mice, only atrophy of the endometrial stromal cells was observed, while the columnar epithelial cells were dilated. Because oestrogen primes the endometrium and plays a key role in inducing its proliferation, the atrophy of the endometrium caused by BHPF once again demonstrated the strong anti-oestrogenicity of BHPF *in vivo*. The weights of the ovaries on both sides were not affected in the BHPF groups, but they were significantly decreased in the 1.2 mg kg$^{-1}$ bw 3-d$^{-1}$ TAM group ($P < 0.01$, Fisher's LSD test). Analysis of ovary sections showed that BHPF and TAM retarded follicular development and reduced the numbers of corpora lutea (Fig. 5d).

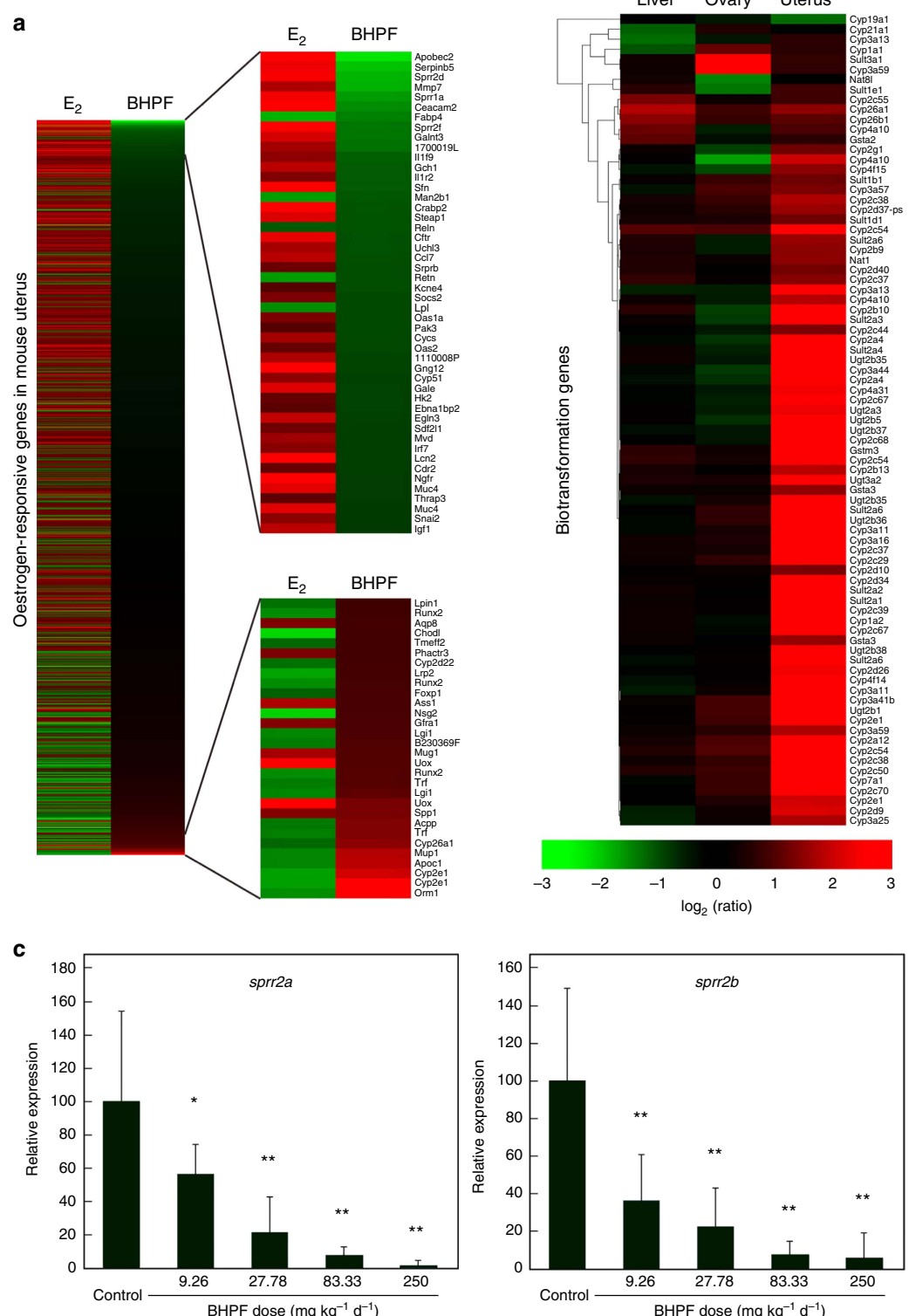

**Figure 4 | Some results of the global gene expression microarrays and the Q-RT-PCR.** (**a**) Comparison of expression profile of oestrogen-responsive genes in the uteri of mice treated with 50 mg kg$^{-1}$ d$^{-1}$ BHPF and that reported in the uteri of mice treated with E$_2$; the fold change indicates the relative expression in the treatment versus the control. (**b**) Hierarchical cluster analysis for the expressions of genes involved in biotransformation in the livers, ovaries, and uteri of mice treated with BHPF, determined by microarray; cluster distances were evaluated by Spearman correlation on the average linkage; the fold change indicates the relative expression in the treatment versus the control. Mice were treated with 50 mg kg$^{-1}$ d$^{-1}$ BHPF for 3 days beginning on PND 24; three mice per group were used for the microarrays. (**c**) Relative expressions of *sprr2a* and *sprr2b* in the uteri of mice given BHPF at doses of 0, 9.26, 27.8, 83.3 and 250 mg kg$^{-1}$ bw d$^{-1}$ for 10 days were determined by Q-RT-PCR; 10 mice per group were used for the assays; the data are expressed as a percentage of the control; error bars indicate the s.d.'s from the average; *$P < 0.05$; **$P < 0.01$; significance was tested by Fisher's LSD test.

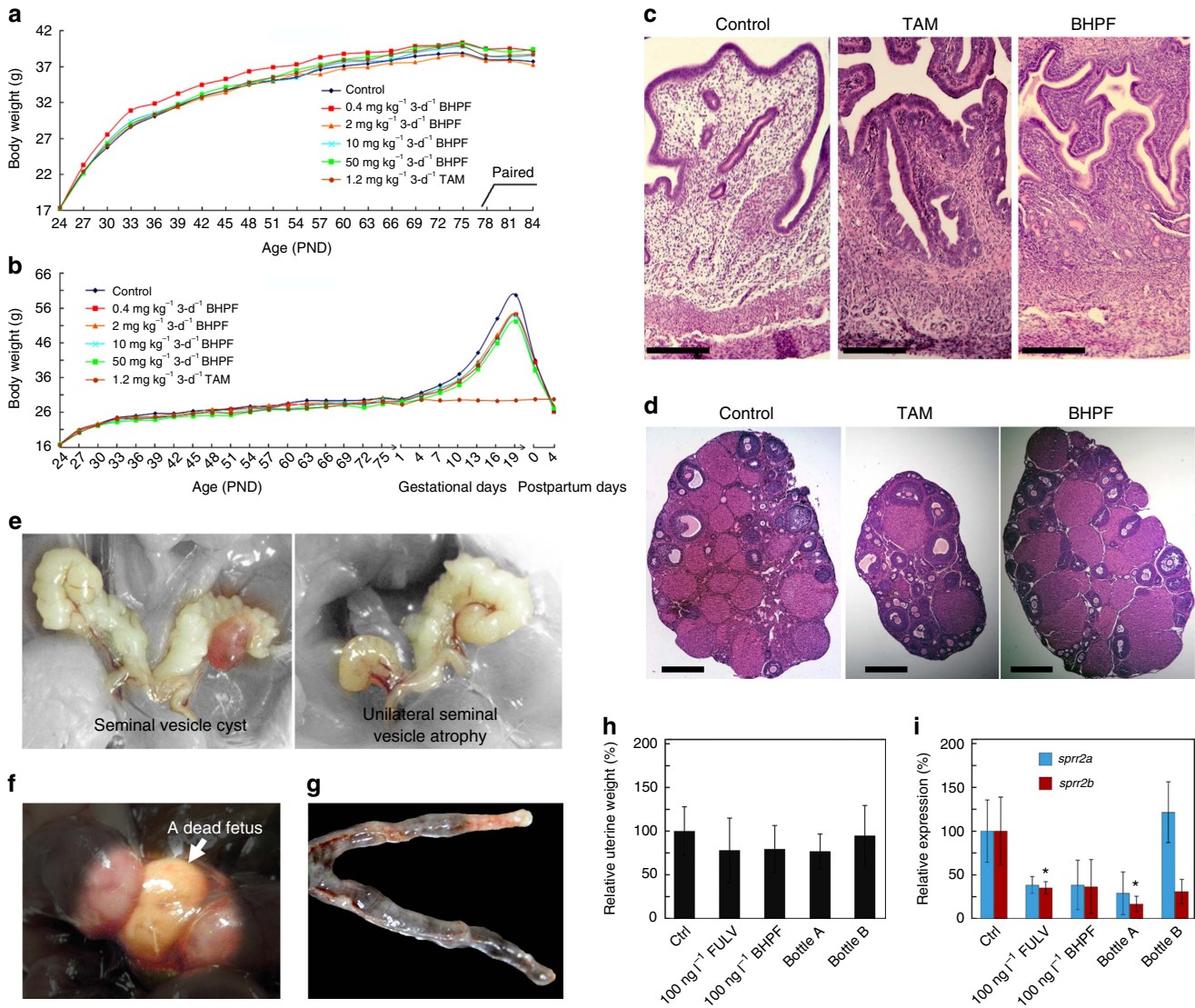

**Figure 5 | Subchronic and reproductive toxicity and low-dose effects of BHPF in mice.** (**a–g**) Subchronic and reproductive toxicity of BHPF. CD-1 mice were treated with TAM or BHPF once every 3 days from PND 24. Body weight evolution of male (**a**) and female (**b**) mice. (**a**) Body weights of male were recorded to PND 84 when a 7-day mating period was just finished; $n = 15$ mice per group. (**b**) Body weights of female were recorded to postpartum days 4; $n = 25$ mice per group before PND 60, and then $n = 15$ mice per group. Photomicrographs of sections of uteri (**c**) and ovaries (**d**) were from mice at PND 60 (H&E staining). The photomicrographs for Control were from mice treated with peanut oil vehicle; the photomicrographs for TAM were from mice treated with TAM at 1.2 mg kg$^{-1}$ bw 3-d$^{-1}$; the photomicrographs for BHPF were from mice treated with BHPF at 2 mg kg$^{-1}$ bw 3-d$^{-1}$. Scale bar, 200 μm in the photomicrographs of uteri; scale bar, 500 μm in the photomicrographs of ovaries. (**e**) Seminal vesicle cyst observed in a mouse from 1.2 mg kg$^{-1}$ bw 3-d$^{-1}$ TAM group and unilateral seminal vesicle atrophy observed in a mouse from 50 mg kg$^{-1}$ bw 3-d$^{-1}$ BHPF group. (**f**) Dead fetus in late pregnancy observed in uterus of a mouse from 10 mg kg$^{-1}$ bw 3-d$^{-1}$ BHPF group. (**g**) A uterus containing insufficiently absorbed embryos observed in a mouse from 50 mg kg$^{-1}$ bw 3-d$^{-1}$ BHPF group. (**h,i**) Low-dose effects of BHPF in mice. Female CD-1 mice were exposed to BHPF, artificially added (100 ng l$^{-1}$ BHPF) or released from 'BPA-free' plastic water bottles, through drinking water for 10 days beginning on PND 24; $n = 8$ mice per group. The levels of BHPF in the water released from Bottles A and B were determined to be 124.35 and 23.81 ng l$^{-1}$, respectively. After the exposure, relative uterine weights (**h**) and relative expressions (**i**) of *sprr2a* and *sprr2b* in the uteri of the mice were determined; the data are expressed as a percentage of the control. Error bars indicate the s.d.'s from the average. *$P < 0.05$, Fisher's LSD test.

After a 7-day mating period, the male mice were weighed and killed at the age of PND 84. The relative weights of the liver, kidney, spleen, testes and accessory sex organs are summarized in Supplementary Table 6. The relative weights of the seminal vesicle and epididymis were found to be significantly ($P < 0.05$, Fisher's LSD test) increased in the 10 mg kg$^{-1}$ bw 3-d$^{-1}$ BHPF group. Pathological examination found one case of seminal vesicle cyst in the 1.2 mg kg$^{-1}$ bw 3-d$^{-1}$ TAM group and one case of unilateral seminal vesicle atrophy in the 50 mg kg$^{-1}$ bw 3-d$^{-1}$ BHPF group (Fig. 5e). No obvious adverse effects were observed in the sperm quality of the BHPF- and TAM-treated mice.

On post-partum day 4, the dams were weighed and killed. The relative uterine weights of the dams were found to be decreased in a dose-dependent manner. The relative uterine weights in the 10 and 50 mg kg$^{-1}$ bw 3-d$^{-1}$ BHPF groups were significantly ($P < 0.05$, Fisher's LSD test) lower than that of the control. Although there were no obvious differences in the number of implantation sites, embryonic resorption and insufficient embryonic absorption could be observed in some

**Table 1 | The main parameters affected by BHPF treatment in the subchronic and reproductive Toxicity test using mice.**

| Endpoints | Ctrl | $0.4 \, \mathrm{mg \, kg^{-1} 3\text{-}d^{-1}}$ | $2 \, \mathrm{mg \, kg^{-1} 3\text{-}d^{-1}}$ | $10 \, \mathrm{mg \, kg^{-1} 3\text{-}d^{-1}}$ | $50 \, \mathrm{mg \, kg^{-1} 3\text{-}d^{-1}}$ |
|---|---|---|---|---|---|
| Relative uterine weight (%) of female mice at PND 60 ($n = 10$) | $0.59 \pm 0.13$ | $0.47 \pm 0.15$ | $0.44 \pm 0.18$* | $0.40 \pm 0.10^\dagger$ | $0.38 \pm 0.06^\dagger$ |
| Number of dams | 15 | 14 | 12 | 11 | 11 |
| Relative uterine weight (%) of dams | $0.55 \pm 0.07$ | $0.54 \pm 0.17$ | $0.52 \pm 0.11$ | $0.46 \pm 0.06$* | $0.43 \pm 0.07^\dagger$ |
| Implantation sites in the uterus of dams | $15.27 \pm 2.18$ | $15.21 \pm 2.02$ | $15.42 \pm 2.02$ | $16.09 \pm 1.81$ | $14.45 \pm 2.42$ |
| Absorption sites in the uterus of dams | 0 | $0.14 \pm 0.36$ | $0.33 \pm 0.65$ | $0.64 \pm 0.67$* | $1.09 \pm 1.38^\dagger$ |
| Number of live pups per litter | $13.60 \pm 2.44$ | $12.71 \pm 2.55$ | $11.67 \pm 2.06$* | $12.73 \pm 1.85$ | $10.27 \pm 2.53^\dagger$ |
| Birth weight of pups | $1.70 \pm 0.18$ | $1.66 \pm 0.21$ | $1.59 \pm 0.24^\dagger$ | $1.54 \pm 0.18^\dagger$ | $1.64 \pm 0.16^\dagger$ |
| Body weight of female pups at PND 4 | $2.82 \pm 0.59$ | $3.08 \pm 0.57^\dagger$ | $3.04 \pm 0.62$* | $3.05 \pm 0.44^\dagger$ | $3.20 \pm 0.53^\dagger$ |
| Body weight of male pups at PND 4 | $3.01 \pm 0.62$ | $3.30 \pm 0.56^\dagger$ | $3.18 \pm 0.59$ | $3.16 \pm 0.49$ | $3.21 \pm 0.51$* |

Relative organ weights are presented as the percentage of organ weight to body weight. Data are expressed as mean ± s.d.
*$P < 0.05$ compared with corresponding control.
†$P < 0.01$ compared with corresponding control. Significance was tested by Fisher's LSD test.

BHPF-administered animals, and the numbers of absorption sites were increased in a dose-dependent manner. Correspondingly, the numbers of live pups per litter in all of the BHPF groups were lower than that of the control, and they were significantly decreased in the 2 and $50 \, \mathrm{mg \, kg^{-1}}$ bw 3-d⁻¹ BHPF groups ($P < 0.05$, Fisher's LSD test). In addition, the birth weights of the pups in the 2, 10 and $50 \, \mathrm{mg \, kg^{-1}}$ bw 3-d⁻¹ BHPF groups were significantly ($P < 0.01$, Fisher's LSD test) lower than that of the control. However, the weights of the pups in all of the BHPF groups were higher than that of the control on PND 4. The acceleration in weight gain in the BHPF groups was attributed to the smaller number of pups per litter in those groups, which therefore received more milk per capita. There were 1, 3, 3 and 2 non-pregnant females in the 0.4, 2, 10 and $50 \, \mathrm{mg \, kg^{-1}}$ bw 3-d⁻¹ BHPF groups, respectively. Histopathological examination found that haemorrhagic ovary cysts occurred in both the TAM and BHPF groups, which may have been another cause of the reduced pregnancy rates besides the atrophic endometria and retarded follicular development. Additionally, in a pregnant female of the $10 \, \mathrm{mg \, kg^{-1}}$ bw 3-d⁻¹ BHPF group that did not parturite 4 days after the estimated date of parturition, a dead fetus in late pregnancy was observed in the uterus (Fig. 5f); and, moreover, one case of late abortion with the uterus containing insufficiently absorbed embryos (Fig. 5g) and one unexplained maternal death during parturition were observed in the $50 \, \mathrm{mg \, kg^{-1}}$ bw 3-d⁻¹ BHPF group.

**Effects of low doses of BHPF relevant to human exposure.** To study the effects of BHPF at doses relevant to human exposure, water samples with BHPF artificially added ($100 \, \mathrm{ng \, l^{-1}}$ BHPF) or released from 'BPA-free' plastic water bottles (Bottle A and Bottle B) were given *ad libitum* to female CD-1 mice beginning on PND 24. The mice of the Bottle A and Bottle B groups received cooled boiled water that had been filled into Bottle A and Bottle B, respectively, while still boiling, and their BHPF levels were determined to be 124.35 and $23.81 \, \mathrm{ng \, l^{-1}}$, respectively. After a 10-day exposure, the relative uterine weights in the groups of $100 \, \mathrm{ng \, l^{-1}}$ BHPF, Bottle A, Bottle B, and the positive control ($100 \, \mathrm{ng \, l^{-1}}$ FULV) were decreased to 79.34% ± 26.99%, 76.62% ± 19.97%, 94.81% ± 34.45% and 77.89% ± 37.50% that of the control ($P > 0.05$, Fisher's LSD test), respectively (Fig. 5h). The gene expressions of *sprr2a* and *sprr2b* in the uteri of the mice were also studied by Q-RT-PCR (Fig. 5i). The expressions of *sprr2a* were decreased in the groups of $100 \, \mathrm{ng \, l^{-1}}$ FULV, $100 \, \mathrm{ng \, l^{-1}}$ BHPF, and Bottle A, but no statistically significant difference was observed ($P > 0.05$, Fisher's LSD test). The gene expressions of *sprr2b* were decreased in all of the test groups, and were significantly ($P < 0.05$, Fisher's LSD test) lower than that of

the control in the groups of $100 \, \mathrm{ng \, l^{-1}}$ FULV and Bottle A. Finally, the serum levels of BHPF in each mouse were determined after enzymatic hydrolysis using β-glucuronidase/arylsulfatase. Serum BHPF was detected only in the mice of the Bottle A group ($1.21 \pm 1.11 \, \mathrm{ng \, ml^{-1}}$), with BHPF levels in the range of $0.36$–$2.70 \, \mathrm{ng \, ml^{-1}}$, but no serum BHPF was detected in the $100 \, \mathrm{ng \, l^{-1}}$ BHPF group.

**Discussion**

Over the past two decades, oestrogenic substances have caused great concern as endocrine-disrupting chemicals[2,3,14–16]. A large number of natural or man-made oestrogenic substances have been listed as environmental oestrogens, while relatively few strongly anti-oestrogenic substances have been reported other than anti-oestrogenic drugs, such as TAM, OHT, raloxifene and FULV. In this study, BHPF showed strong anti-oestrogenic activity in a yeast two-hybrid assay and dual-luciferase reporter assays (Fig. 2b,d,f). The uterotrophic and vaginal cell cornification assays are the most widely used *in vivo* assays for assessing oestrogenic substances[19]. Although the vaginal cell cornification assay has the advantage of being relatively simple and can use the same animals repeatedly if the test compound does not bioaccumulate, it is rarely used to evaluate anti-oestrogenic substance and it has been criticized as being largely qualitative as scoring is dependent on the evaluation of cellular contents of a vaginal lavage[19]. The uterotrophic assay has been served as a test for oestrogenic and anti-oestrogenic substances since 1930s (ref. 20). Previous studies examining the effects of oestrogenic and anti-oestrogenic substances on uterine wet weight have used a number of different protocols and species; therefore, uterotrophic assay requires validated operating procedures that specify species, strain, age and route of test compound administration[19,20]. In this study, we validated the uterotrophic assay using FULV, a full oestrogen receptor antagonist, and then evaluated the anti-oestrogenicity of BHPF. We found that BHPF significantly inhibited the uterine weights of CD-1 mice in a dose-dependent manner, and the anti-uterotrophic activity of BHPF at a dose of $27.8 \, \mathrm{mg \, kg^{-1}}$ bw d⁻¹ was close to that of FULV at $2 \, \mathrm{mg \, kg^{-1}}$ bw d⁻¹ (Fig. 3c,d). We analysed the global gene expression profiles in the livers, ovaries and uteri of mice treated with $50 \, \mathrm{mg \, kg^{-1}}$ d⁻¹ BHPF for 3 days. Gene ontology analysis of the entire sets of up-regulated or down-regulated genes in the ovaries and uteri of mice treated with BHPF revealed the biological processes in response to $E_2$ stimuli. By comparing the gene expression profile in the uteri of mice given BHPF with that reported in the uteri of mice given $E_2$ (ref. 18), we found that most of the oestrogen-responsive genes exhibited the opposite expression pattern in the uteri of mice treated with BHPF to that

induced by $E_2$ (Fig. 4a). Of particular note is the down-regulation of SPRRs, a cluster of genes known to be induced by oestrogens through nuclear oestrogen receptors in mouse uteri[21]. Moreover, it should be noted that BHPF might induce anti-oestrogenic effects through mechanisms other than nuclear oestrogen receptors, as reported for BPA and some other bisphenols, which can induce adverse effects through mechanisms including the oestrogen membrane receptor, oestrogen-related receptor gamma, pregnane X receptor and so on[2,16,22–24]. Our microarray analysis also showed that some genes involved in biotransformation and oestrogen metabolism were up-regulated in the uteri of the BHPF-treated mice (Fig. 4b), and the up-regulation of these genes may facilitate the in utero metabolism of oestrogen, thereby suppressing the effect of oestrogen on the uterus. Molecular docking of BHPF into the antagonist pocket of ER$\alpha$ (PDB ID 3ERT) showed that BHPF could be well accommodated by this pocket (Fig. 2h). Interestingly, even though BHPF could not be accommodated by the bisphenol C (BPC) pocket of the oestrogen receptor $\alpha$ structure (PDB ID 3UUC), the optimal position of BHPF in oestrogen receptor $\alpha$ (PDB ID 3ERT) was very similar to that of BPC in the oestrogen receptor $\alpha$ structure (3UUC)[25]. It was previously reported that BPC displayed almost full antagonistic activity in the presence of $E_2$, and that oestrogen receptor $\alpha$ with BPC displayed an antagonist conformation similar to that of the OHT-bound structure (3ERT)[25]. In addition, it was found that BHPF inhibited the proliferation of MCF-7 human breast cancer cells in a granted patent (CN102727470) (ref. 26). These findings indicated the strong anti-oestrogenic activity of BHPF.

Serious developmental and reproductive effects of BHPF associated with its anti-oestrogenic activity were observed in this study. In the subchronic and reproductive toxicity tests, BHPF significantly decreased the uterine weights and induced atrophic endometria in the intact female mice at doses of 2 mg kg$^{-1}$ 3-d$^{-1}$ or higher by oral gavage for 36 days from PND 24 (Table 1). These findings are consistent with a previous study on the anti-oestrogen EM-800, which was found to reduce uterine weights and cause atrophy of the endometrium following 28 days of oral administration in intact female BALB/c mice at an age of $\sim$50 days[27]. We also found that BHPF reduced pregnancy-related weight gain and caused embryonic absorption and fetal death during pregnancy, and caused decreased birth weights of the pups, in the reproductive toxicity study (Table 1). Because uterine stromal cells are known to play a critical role in pregnancy-associated neovascularization and embryo survival[28], the adverse pregnancy outcomes were thought to have been caused by the atrophic endometria. Kaplan-Kraicer et al.[17] orally administered TAM or the anti-oestrogen RU39411 to rats before implantation on day 2 of pregnancy, and found that low doses of TAM and RU39411 reduced the litter sizes and weights and induced embryonic absorption in some animals, while high doses prevented pregnancy in all animals. The reproductive effects of BHPF observed in this study are similar to those of RU39411 and TAM, suggesting that its anti-oestrogenicity plays a critical role in its reproductive toxicity. The incidence of adverse pregnancy outcomes, such as spontaneous abortion, preterm delivery, low birth weight and fetal death in humans, has increased in many countries[29–31], and it has been proposed that exposure to certain environmental pollutants may be linked with these adverse pregnancy outcomes[16,30,31]. This study provided evidence that anti-oestrogenic chemical exposure may cause adverse pregnancy outcomes, suggesting that environmental anti-oestrogens, as well as their adverse effects on human reproductive health, should be of concern.

In this study, we recruited 100 college student volunteers (50 female and 50 male) who habitually drank from plastic bottles, and measured their serum BHPF levels by GC–MS. Although we do not know whether these bottles contained BHPF, we found that BHPF was detectable in 7 volunteers (4 male and 3 female), with a mean serum BHPF level of $0.34 \pm 0.21$ ng ml$^{-1}$ and a maximal level of 0.7 ng ml$^{-1}$. When we administered cooled boiled water with a concentration of 124.35 ng l$^{-1}$ of BHPF released from plastic bottles, ad libitum to female CD-1 mice for 10 days, beginning on PND 24, we detected serum BHPF levels of $1.21 \pm 1.11$ ng ml$^{-1}$ in the group receiving water from Bottle A, suggesting that drinking cooled boiled water from BHPF-containing plastic bottles may be sufficient to elevate BHPF levels in human blood. In China and some other countries, drinking boiled water is routine. Plastic bottles, being easily transportable, are often used to store boiled drinking water. Student populations (from elementary students to college students) are among the most common users of plastic bottles. Many students carry plastic bottles every day, and mostly drink boiled water collected from public drinking-water heaters. However, because students (particularly elementary and middle school students) are at developmental stages from prepuberty to sexual maturity, the effects of anti-oestrogenic pollutants on these populations deserve particular attention.

In the dual-luciferase reporter assays, the IC$_{50}$ values of BHPF, that is, the concentrations of BHPF inhibiting 50% of the activity of 1 nM ($\approx$272.38 pg ml$^{-1}$) $E_2$, were $1.09 \times 10^{-7}$ M for oestrogen receptor $\alpha$ and $7.53 \times 10^{-8}$ M for oestrogen receptor $\beta$ (Fig. 2d,f); converting the molar concentrations to mass concentrations, these correspond to 38.19 and 26.39 ng ml$^{-1}$, respectively. Therefore, the IC$_{50}$ values of BHPF were only $\sim$54.56 and 37.70 times higher than the maximal serum BHPF level (0.7 ng ml$^{-1}$) detected in the human volunteers, and $\sim$31.56 and 21.81 times higher than the mean serum BHPF level of $1.21 \pm 1.11$ ng ml$^{-1}$ in the female mice of the Bottle A group. It is notable that 272.38 pg ml$^{-1}$ $E_2$ is a relatively high level in serum for healthy adult women, which would be expected to occur only as the follicle matures, with peak $E_2$ levels of $\sim$200–300 pg ml$^{-1}$ just before ovulation in the normal menstrual cycle. Moreover, in the female mice given water containing BHPF at concentrations relevant to human exposure, low uterine weights and decreased expressions of oestrogen-responsive genes were observed (Fig. 5h,i). These results indicated the risk that BHPF may antagonize endogenous oestrogens and disrupt the normal physiological effects of oestrogens at doses relevant to human exposure.

In principle, substitutes for hazardous chemicals should present lower risks than the original substances. Several guideline-compliant toxicity studies have found a systemic no-observed-adverse-effect level (NOAEL) of 5 mg kg$^{-1}$ bw day$^{-1}$ for BPA and a reproductive/developmental NOAEL of 50 mg kg$^{-1}$ bw day$^{-1}$ for BPA in rats and mice[32–34]. However, some BPA substitutes, such as bisphenol S, bisphenol F, bisphenol AF and BPC, were recently reported to be as hormonally active as BPA, and have been detected in food and human urine samples in several countries[25,35–38]. In this study, we found that BHPF could be released from plastic bottles into drinking water, whereupon it was detectable in the serum samples of a small proportion of human volunteers; it induced atrophic endometria at doses of 0.4 mg kg$^{-1}$ bw 3-d$^{-1}$ or higher and caused adverse pregnancy outcomes at doses of 2 mg kg$^{-1}$ bw 3-d$^{-1}$ or higher in CD-1 mice; and in female mice given water containing BHPF at doses relevant to human exposure, low uterine weights and decreased expressions of oestrogen-responsive genes were observed. These findings suggest that substitutes for BPA may also be toxicologically problematic.

In vitro and in vivo study demonstrated that BHPF was strongly anti-oestrogenic. Subchronic and reproductive toxicity

tests using CD-1 mice showed that BHPF could reduce uterine weights and induce atrophic endometria in females, reduce pregnancy-related weight gain and cause embryonic absorption and fetal death during pregnancy, and reduce the birth weights of pups at doses lower than the NOAEL reported for BPA. In prepubertal female mice given *ad libitum* water containing BHPF at concentrations relevant to human exposure, low uterine weights and decreased expressions of oestrogen-responsive genes were observed; and BHPF was detected in the serum of mice given cooled boiled water containing higher levels of BHPF released from plastic bottles. These results hint at potential the health risks of BHPF, suggesting that it may not be safe for the use in materials that come into contact with food. Screening programs for endocrine disruptors have been established in many countries in recent years, but many of these programs do not screen compounds for anti-oestrogenic activity. Our results suggest that anti-oestrogenic pollutants, such as BHPF, as well as their adverse effects on human health, warrant further study. Moreover, this study raises questions about the safety of BPA substitutes and the current toxicological management of substitutes for hazardous chemicals.

## Methods

**Chemicals.** Methanol, acetone and hexane were HPLC-grade, obtained from Fisher Chemicals (Fair Lawn, NJ). Dimethyl sulfoxide (DMSO) was ACS-grade, obtained from AMRESCO. BHPF ($>98\%$), $E_2$ ($\geq 98\%$), OHT ($\geq 98\%$), FULV ($>98\%$) and 9-fluorenone ($>98\%$) were purchased from Sigma-Aldrich (St Louis, MO). TAM ($\geq 99\%$) was obtained from Aladdin Reagents (Shanghai, China). Phenol-$d_5$ (RING-D5, $>98\%$) was purchased from Cambridge Isotope Laboratories (Andover, MA). β-Mercaptopropionic acid ($>98\%$) was obtained from Tokyo Chemical Industry Co., Ltd. (Tokyo, Japan).

**Identification of BHPF from a plastic bottle by NMR.** Methanol leachates from the grated plastic of a 'BPA-free' bottle were isolated and purified by liquid chromatography with a C18 preparative column. The purified substance was used for $^1$H and $^{13}$C NMR analysis. $^1$H NMR and $^{13}$C NMR spectra were recorded at 400 and 600 MHz on a Bruker AVANCE 400 spectrometer and a Bruker AVANCE DRX-600 spectrometer, respectively.

**Synthesis of deuterated BHPF.** A mixture of 9-fluorenone, phenol-$d_5$ and β-mercaptopropionic acid was added to a 50 ml three-neck bottom flask equipped with a magnetic stirrer, reflux condenser and thermometer, and then concentrated sulphuric acid was slowly added dropwise with stirring. The mixture was heated at 55 °C for 4 h, and then washed several times with water and methanol, and dried. The dried powder was recrystallized from toluene, and deuterated BHPF was obtained. By analysis of this deuterated BHPF, it was identified as BHPF-$d_7$ (Supplementary Fig. 1a,b), which was used as a surrogate standard for the GC–MS analysis of BHPF.

**Analysis of BHPF in water released from plastic bottles.** Fifty two plastic bottles, including adult water bottles, children water bottles and sippy cups, and baby milk bottles, with volume in the range of 240–1,000 ml, were purchased in China and in foreign markets (Supplementary Table 1). The plastic bottles were filled with water heated to 60 °C, and allowed to sit for 6 h before the determination of BHPF. A water sample spiked with 20 ng of surrogate standard (BHPF-$d_7$) was extracted on a Phenomenex strata-x SPE cartridge (500 mg 6 ml$^{-1}$). The cartridges were preconditioned with 20 ml of methanol and 8 ml of distilled water, and then rinsed with 5 ml of distilled water and 8 ml of methanol/water (1:20, v/v). After being dried under a flow of nitrogen, the cartridge was eluted with 8 ml of methanol. The extracts were then dried under a flow of nitrogen and derivatized with 100 μl BSTFA + 1% TMCS (Regis Technologies, Inc., Morton Grove, IL) in acetone. After 30 min of reaction, 1 ml of water was added to the incubation mixtures, and they were extracted with *n*-hexane (1:1) three times. The extracts were evaporated to dryness under a gentle nitrogen stream and dissolved in 100 μl of *n*-hexane for GC–MS analysis (Agilent 7890A/5975C). An HP-5MS capillary column (30 m × 0.25 mm × 0.25 μm film thickness; Agilent) was used to separate the target chemicals. A splitless injector was used and was maintained at 300 °C. The temperature program ranged from 100 to 300 °C at a rate of 15 °C min$^{-1}$. The interface temperature, ion temperature and quadrupole temperature were maintained at 300, 230 and 150 °C, respectively. The carrier gas was helium supplied at a constant flow rate of 1 ml min$^{-1}$. The injection volume was 1 μl. As shown in Supplementary Fig. 1b, *O*-bis(trimethylsilyl) derivatives were observed at $m/z = 494$ and 501 for BHPF and BHPF-$d_7$, respectively. The recoveries of BHPF

and BHPF-$d_7$ in the spiked water samples were in the range of 80.8–103.5%, and the limit of detection was calculated as 0.1 ng l$^{-1}$.

**Analysis of BHPF in human serum.** In China, college students usually undergo a physical examination before graduation. The volunteers in this study were senior students who were undergoing a physical examination in Beijing. Fifty male and 50 female healthy volunteers (mean age, 23.5 ± 1.2 years) who habitually used plastic water bottles were randomly recruited to participate in this study in June 2015, after appropriate ethical approval was received from the Human Ethics Committee of the Peking University Peoples' Hospital. Each participant provided informed consent and volunteered to give one 10-ml random sample of blood. Fasting blood was collected at Peking University Hospital Medical Center with vacutainer blood collection device (Chengwu Yongkang Medical Products Ltd., China) between 8 and 9 h. The blood collection device principally consisted of a steel needle with sharp ends, which linked the blood–vessel lumen with the glass vacutainer tube during blood collection, and a polypropylene auxiliary syringe for holding the blood, which had been verified to be free of BHPF contamination. The blood was collected directly into plain 10-ml glass vacutainer tubes and spun down 30 min later for serum preparation. The serum samples were stored in glass tubes at − 80 °C until analysis. The samples were fortified with 5 ng of surrogate standard BHPF-$d_7$, and the pH was adjusted to 6.5 with ammonium acetate. β-Glucuronidase/arylsulfatase (20 μl; Sigma-Aldrich, St Louis, MO) was added to the samples, and they were incubated at 37 °C for 20 h to allow deglucuronidation. After enzymatic hydrolysis, the samples were diluted and loaded onto a Phenomenex strata-x SPE cartridge (200 mg 3 ml$^{-1}$) that had previously been conditioned with 6 ml of methanol and 3 ml of distilled water. The cartridge was rinsed with 3 ml of distilled water and 4 ml of methanol/water (1:20, v/v). After being dried under a flow of nitrogen, the cartridge was eluted with 4 ml of methanol. The eluate was evaporated to dryness, and dissolved in *n*-hexane for GC–MS analysis. All analytical procedures were checked for precision, reproducibility, blank contamination and linearity. Quality control was maintained by analysing a method blank (calf serum) and two spiked calf serum samples (piked with BHPF-$d_7$ singly or a mixture of BHPF-$d_7$ and undeuterated BHPF) along with every 12 samples. No BHPF was detected in the blank and single BHPF-$d_7$ spiked calf serum samples during the analytical procedures. The detection limit (0.1 ng ml$^{-1}$) was based upon the criterion of the instrument having a signal-to-noise response of 3:1.

**Yeast two-hybrid assay.** The oestrogenic and anti-oestrogenic activities of BHPF were tested by a yeast two-hybrid assay with human oestrogen receptor α and the coactivator transcriptional intermediary factor 2 (refs 39,40). Yeast cells (*Saccharomyces cerevisiae* Y190) expressing the human oestrogen receptor α (ERα) were grown overnight at 30 °C with vigorous shaking in synthetic defined medium lacking tryptophan and leucine. The overnight culture was then added to fresh medium and treated with test chemicals at 30 °C for 4 h. After the incubation, the treated cells were collected and digested with 1 mg ml$^{-1}$ Zymolyase-20T (Seikagaku Kogyo Co., Ltd., Tokyo) at 37 °C for 20 min, and then 2-Nitrophenyl β-D-galactopyranoside was added to the lysate to a final concentration of 4 mg ml$^{-1}$. After incubation at room temperature for 20 min, the yellow colour developed and 100 ml of 1 M Na$_2$CO$_3$ were added to stop the reaction. Then, the yeast debris was removed by centrifugation and β-galactosidase activities were determined by measuring the absorbance of supernatant at 415 nm in a spectrophotometer. To determine whether BHPF possessed anti-oestrogenic activity, 1 nM of $E_2$ with the diluted BHPF was added to the medium. OHT was used as a positive control. All experiments were performed in triplicate. Sigmoidal dose–effect curves were calculated using the GraphPad Prism 4 software. The IC$_{50}$ values were calculated on the basis of the sigmoidal dose-effect curves.

**Dual-luciferase reporter assays.** Dual-luciferase reporter assays were carried out primarily according to the method described in a previous study[41]. In short, full-length sequences of human oestrogen receptor α and β were amplified by RT-PCR using mRNA from MCF-7 (Supplementary Table 3) and inserted into a pSVSPORT1 (Invitrogen); the resulting constructs were termed pSVhERα and pSVhERβ. A reporter plasmid containing ERE was constructed by inserting four tandem repeats of an ERE of *Xenopus laevis* vitellogenin A2 into a pGL4.23 vector (Promega), and was termed pGL4-ERE-luc. All of the plasmids constructed were confirmed by sequence analysis. The Renilla LUC control reporter construct pGL4.74-TK was purchased from Promega (Madison, WI). Human choriocarcinoma JEG-3 cells (ATCC No. HTB-36) were obtained from the American Type Culture Collection (ATCC; Manassas, VA) and were cultured in minimal essential medium (MEM, Invitrogen, Carlsbad, CA) containing 2 mM L-glutamine, 0.1 mM minimal essential medium nonessential amino acid solution (Invitrogen/Thermo Fisher Scientific, Grand Island, NY), and 10% fetal calf serum (FCS) at 37 °C in a humidified atmosphere containing 5% CO$_2$. The cells (3 × 10$^4$ cells per well) were seeded in 24-well plates 24 h before transfection. The cells were transfected with pGL4-ERE-luc (5 ng per well), pGL 4.74-TK (0.2 ng per well) and either pSVhERα (5 ng per well) or pSVhERβ (5 ng per well) using Lipofectamine reagent (Invitrogen). At 24 h after transfection, the compounds in DMSO were added to the cultures at a volume ratio (v/v) of 0.1%. The cells were

continuously cultured in medium supplemented with 1% charcoal-stripped FCS for another 24 h and then harvested, and cell extracts were assayed for firefly luciferase activity. The assays were performed in quadruplicate, and the results were expressed as the fold induction of the control after normalization against Renilla luciferase activity. Sigmoidal dose-effect curves were calculated using the GraphPad Prism 4 software. The $IC_{50}$ values were calculated on the basis of the sigmoidal dose-effect curves. Finally, a WST-1 cell proliferation assay was performed to evaluate the cytotoxicity of BHPF using the cell proliferation reagent WST-1 (Dojindo, Mashiki, Japan). Aliquots (200 μl) of JEG-3 cells ($1 \times 10^4$ cells per well) were seeded into 96 well-plates and precultured for 24 h. The cells were then treated with various concentrations of BHPF. After another 24 h, 20 μl of 5 mM WST-1 with 0.2 mM 1-Methoxy-5-methylphenazinium methylsulfate was added for 4 h before the absorbance at 490 nm was measured with a microplate reader (iMark, BioRad). The WST-1 cell proliferation assay was performed in quadruplicate.

**Molecular docking.** Scigress (Ultra Version 3.0.0, Fujitsu) was used for *in silico* molecular docking analysis[42,43]. The three-dimensional structures of the ligand-binding domain of human oestrogen receptor α, PDB IDs 1ERE, 3UU7, 3UUA, 3UUC and 3ERT, were downloaded from the Protein Data Bank website (http://www.rcsb.org.pdb) and used to evaluate the binding affinities of BHPF in the agonist and antagonist pockets of human oestrogen receptor α, respectively. The structures were refined, and all of the water molecules were removed from the protein, except for those that were important to the ligand-binding pocket composition, using Scigress-integrated procedures. The structures of BHPF and the original ligands were drawn, cleaned, and energy-optimized for molecular modulation. Automated docking of the flexible ligands into the flexible active sites was carried out using a genetic algorithm. The docking calculations were evaluated on a 0.25 Å grid in a $15 \times 15 \times 15$ Å box containing the active site for the original ligand. The local search parameters included 300 maximum iterations at a rate of 0.06, and the procedure was set to run 30,000 generations with an initial population size of 50, an elitism of 7, a crossover of 0.8, a mutation of 0.2, and a convergence of 1.0.0. After automated docking, a geometric optimization calculation was performed for the human oestrogen receptor α-BHPF complex using augmented MM3 parameters.

**Anti-uterotrophic assays using mice.** All animal studies were approved by the Institutional Animal Care and Use Committee of Peking University, and were performed in accordance with the Guidelines for Animal Experiments of the university, which meet the ethical guidelines for experimental animals in China. Immature female CD-1 mice (20 days old) were obtained from Experimental Animal Tech Co. of Weitonglihua (Beijing, China). Mice with a maximal difference in body weight of 1 g were selected for the experiments and randomly assigned to either the treatment or control groups. The animals were housed five to a cage and acclimatized in a controlled environment with a temperature of $22 \pm 2$ °C, a relative humidity between 40 and 60%, and artificial lighting in a 12 h/12 h light-dark cycle. The animals were fed *ad libitum* with a basic diet from the Laboratory Animal Center of the Academy of Military Medical Sciences (Beijing, China), and drinking water was provided *ad libitum*. Before the experiments, the mice were randomly assigned to either the treatment or the control group ($n = 10$). Mice treated with peanut oil were used as a vehicle control, and the chemicals were dissolved in peanut oil to the appropriate concentrations. The volume of vehicle or chemical solutions administered was adjusted daily based on body weight measured during the dosing period according to the volume-body weight ratio of 5 ml kg$^{-1}$ bw. The treatment via oral gavage to each mouse was performed for 10 days beginning on PND 24. Mice in each cage were labelled by shaving hair on different parts of the body, and the oral gavage treatment was performed in turns for each group, whereby in each turn only one mouse in a group was treated. After the period of treatment, the mice were weighed and killed 24 h after the final treatment according to the sequence of the treatments, and the uteri were removed, blotted and weighed by one experimenter to reduce the risk of bias in the data collection. The relative uterine weight, that is, the ratio of uterine weight to final body weight, was calculated to evaluate the anti-uterotrophic activity of the chemicals.

**Expression profiling and Q-RT-PCR.** Immature female CD-1 mice were obtained from Experimental Animal Tech Co. of Weitonglihua (Beijing, China) and acclimatized in an experimental environment with a temperature of $22 \pm 2$ °C, relative humidity between 40 and 60%, and artificial lighting in a 12 h/12 h light-dark cycle. The animals were fed *ad libitum* with a basic diet from the Laboratory Animal Center of the Academy of Military Medical Sciences (Beijing, China), and drinking water was provided *ad libitum*. The mice were treated with 50 mg kg$^{-1}$ bw d$^{-1}$ of BHPF or peanut oil via oral gavage for 3 days beginning on PND 21. On PND 24, mice of each group were weighed and killed by cervical dislocation. The uteri, ovaries and livers were collected for total RNA isolation. Total RNA was isolated using Trizol reagent (Invitrogen) and further purified by NucleoSpin RNA Clean-up (Macherey-Nagel, Germany). Fluorescence labelling of the RNA samples was performed with a Jingxin cRNA linear amplification and labelling kit (CapitalBio) according to the manufacturer's instructions. The total RNA from each pool of tissue samples (a mixture from three animals) was used for microarray

analysis. Cy3 dCTP was used to label the samples. The labelled products were added to SurePrint G3 mouse gene expression $8 \times 60$ K microarrays (Agilent) and incubated overnight at 42 °C. Following hybridization, the gene arrays were washed and scanned using an Agilent G2565CA Microarray Scanner. Glyceraldehyde 3-phosphate dehydrogenase (GAPDH) was used as an endogenous control to monitor the quality of the target preparation. The data were processed with feature extraction software (GeneSpring GX v.11; Agilent). The raw signals were log-transformed and normalized with the percentile shift normalization method. All arrays met the minimum Agilent QA/QC standards. To verify the data from the gene array, 14 genes were chosen as candidates for Q-RT-PCR. Primers are listed in Supplementary Table 4. GAPDH was used as an endogenous control. Amplification was performed using SYBR Green PCR master mix (Applied Biosystems) according to the manufacturer's instructions. The relative expression was quantified using the $2^{(-\Delta\Delta Ct)}$ method. The Gene Ontology (GO) and KEGG pathway enrichment analyses were performed with DAVID (refs 40,41). Sprr2a and sprr2b were selected as biomarker genes, and their expressions in the uteri of the mice ($n = 10$) given BHPF at doses of 0, 9.26, 27.8, 83.3 and 250 mg kg$^{-1}$ bw d$^{-1}$ for 10 days were studied by Q-RT-PCR.

**Subchronic and reproductive toxicity tests using mice.** Immature CD-1 mice (20 days old) were obtained from Experimental Animal Tech Co. of Weitonglihua (Beijing, China) and acclimatized in an experimental environment with a temperature of $22$ °C $\pm 2$ °C, relative humidity between 40 and 60%, and artificial lighting in a 12 h/12 h light-dark cycle. The animals were fed *ad libitum* with a basic diet from the Laboratory Animal Center of the Academy of Military Medical Sciences (Beijing, China), and drinking water was provided *ad libitum*. The animals were randomized into 6 groups (vehicle control group, 0.4, 2, 10 and 50 mg kg$^{-1}$ BHPF groups, and 1.2 mg kg$^{-1}$ TAM group) with 15 males and 25 females per group. The animals were housed five to a cage until the pairing date. The treatment via oral gavage of each mouse was performed once every 3 days beginning on PND 24. The chemicals were dissolved in peanut oil to the appropriate concentrations. The volume of vehicle or chemical solutions administered was adjusted daily based on body weight measured during the dosing period according to the volume-body weight ratio of 5 ml kg$^{-1}$ bw. On PND 60, 10 females of each group were weighed and killed by cervical dislocation. The uteri, ovaries, livers, kidneys and spleens were weighed and collected for histological analysis. At the age of PND 77, males and females were paired. Each morning, the females were examined for the presence of vaginal plugs. Gestational day 1 was defined as the day on which a vaginal plug was found. In cases where pairing was unsuccessful, females were re-mated with proven males of the same group, except that females in the 1.2 mg kg$^{-1}$ TAM group were re-mated with proven males of the control group. After a 7-day mating period, the males were weighed and killed, and organs were weighed and collected for further analysis. Cauda epididymal sperm was collected from each mouse and observed under a trinocular microscope (Olympus BX53) in physiological saline. A microscopic video was recorded and analysed for sperm quality evaluation. The treatment via oral gavage was continued for the females until day 19 of pregnancy. On post-partum days 0 and 4, the weights of the dams and pups were recorded and the pups were checked for abnormalities. On post-partum day 4, the dams were weighed and killed, and their organs were weighed and collected for further analysis. The ovaries and uteri were examined for the numbers of corpora lutea and implantation sites, respectively. Finally, the non-pregnant and nulliparous females were killed for histopathological analysis. For histological examination, organs and tissues were fixed in formalin fixation, embedded in paraffin, and sectioned serially at 5 μm. Slides were stained with hematoxylin and eosin (H&E), and observed under a light microscope (Olympus BX53).

**Exposure experiment through drinking water using mice.** Immature female CD-1 mice (20 days old) were obtained from Experimental Animal Tech Co. of Weitonglihua (Beijing, China). Mice with a maximal difference in body weight of 1 g were selected for the experiments and randomly assigned to either the treatment or control groups. The animals were housed four to a cage and acclimatized in a controlled environment with a temperature of $22$ °C $\pm 2$ °C, relative humidity between 40 and 60%, and artificial lighting in a 12 h/12 h light-dark cycle. The animals were fed *ad libitum* an oestrogen-free diet from Trophic Animal Feed High-tech Co., Ltd. (Nantong, China), and drinking water was provided *ad libitum* in glass bottles. Before the experiments, the mice were randomly assigned to five groups ($n = 8$). Ultra-pure water was used as a control. FULV was dissolved in ultra-pure water to a concentration of 100 ng l$^{-1}$ and used as a positive control. BHPF was dissolved in ultra-pure water to prepare a concentration of 100 ng l$^{-1}$, which is relevant to human exposure. Two plastic drinking bottles labelled 'BPA-free', of different brands, were purchased and denoted Bottle A and Bottle B. The plastic bottles were filled with boiling ultra-pure water from a stainless steel electric water heater and allowed to cool down to room temperature before the animal experiment. The BHPF levels in the cooled boiled waters from Bottle A, Bottle B and the stainless steel electric water heater were determined by GC–MS. No BHPF was detected in the cooled boiled water from the stainless steel electric water heater. The cooled boiled waters from Bottle A and Bottle B were transferred to glass bottles during the experiment. The exposure experiment was performed for 10 days beginning on PND 24. After the period of exposure, the mice were weighed and killed in turns for each group, whereby in each turn only one mouse in a group

was killed. Blood was collected by cardiac puncture soon after each animal was killed, and the uteri were removed, blotted and weighed by one experimenter to reduce the risk of bias in the data collection and immediately frozen in liquid nitrogen for gene expression analysis. The relative uterine weight was calculated to evaluate the anti-uterotrophic activity. The serum was separated by centrifugation and frozen at $-20\,°C$ for BHPF analysis. The serum BHPF levels were analysed by GC–MS using the same method as that for human serum. The gene expressions of *sprr2a* and *sprr2b* in the uteri were determined by Q-RT-PCR.

**Data analysis.** Data analysis was performed with the statistical program SPSS (v.18.0; Chicago, IL). The data are presented as the mean and s.d. unless otherwise indicated. Group differences were assessed by one-way analysis of variance and Fisher's LSD tests after running the one-sample Kolmogorov–Smirnov test and the test of homogeneity of variances. $P$ values of $<0.05$ were considered statistically significant.

**Data availability.** Microarray data are deposited in Gene Expression Omnibus (GEO) database (http://www.ncbi.nlm.nih.gov/geo/) under accession code GSE74066. All the other data supporting the findings of this study are available from the corresponding authors on request.

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

## Acknowledgements

This work was supported by the National Natural Science Foundation of China Grants (grant numbers 21377007, 41330637), the International S&T Cooperation Program of China (2016YFE0117800) and the 111 project (grant numbers B14001).

## Author contributions

Z.Z. and J.H. designed the research; Z.Z., Y.H., J.G., T.Y., L.S., X.X., D.Z., T.N., Y.-h.H., J.-y.L., X.F., S.C., J.L. and X.G. performed the experiments; Z.Z., Y.H., J.G., D.Z., T.N. and Y.-h.H. analysed the data; Z.Z., Y.H., D.Z., T.N., Y.W. and J.H. contributed to the discussion and manuscript; Z.Z., Y.H. and J.H. wrote the paper.

**Additional information**

**Competing financial interests:** The authors declare no competing financial interests.

