## [Peer review file · Nature Communications]

Reviewers' comments:

Reviewer #1 (Remarks to the Author):

A. Summary. This manuscript describes work showing that BHPF is an antiestrogen, that it can leach from some plastic water bottles, that it can be found in human serum. These are important and original observations that have important implications globally.

B. This is highly original and of great interest.

C. Data and methodology. This reviewer cannot evaluate the quality of the chemistry approaches. However, there are potential concerns for consideration. First, the YES assay has certain weaknesses associated with the mosaic nature of the target receptor. It would strengthen the body of work (which is considerable already) to use a full length human receptor and evaluate separate effects on ERa and ERb. Recognizing that this may be deemed to be outside the scope of this original paper. The description of the data in humans should be perhaps more completely described. First, this should not be described as "general population", but rather volunteers. How were these volunteers identified and selected? What age, sex, etc. Other characteristics? This self-selection process could produce a bias. Second, are the authors confident that the measurements of BHPF in human serum is not the result of contamination from plastics employed in the procedure? If no field blanks or tests of equipment was performed, the authors need to be very careful about these conclusions. Moreover, there were very few volunteers that exhibited BHPF levels and this should be reflected in the abstract and summaries ("...was detected in a small proportion of volunteers" ...). The animal studies appear appropriate. However, it might be useful to describe the methods employed to reduce the risk of bias in data collection. Method of animal assignment, sequence of gavage and animal collection, time of day, etc. A separate subsection in the supplementary material might be useful for this.

D. Statistics and Uncertainties. The human data may be the most uncertain and this should be clearly stated.

E. Conclusions are appropriate. However, it may be important to point out that the US EPA does not employ an "antagonist mode" in their estrogen receptor HTS assays in ToxCast because environmental anti-estrogens have not been identified. The work here has very important implications for this.

F. Improvements listed above.

G. References are appropriate

H. Well-written manuscript.

Reviewer #2 (Remarks to the Author):

This study identified a BPA substitute, BHPF, in plastic water bottles, found that it is detectable in the water through leaching at 60C, and tested its potential actions as an endocrine-disrupting chemicals (EDC), particularly as an anti-estrogen through in vitro and in vivo experiments. They also showed that BHPF is detectable in humans. The identification of EDC activity of replacement chemicals is very important, and the study implicates BHPF as another such chemical, alongside other bisphenols such as BPS, BPF, and others. The major strength of work is making comparisons across all of these different levels. There are also novel aspects of work such as the docking experiment. The weakness is that much of the work is done at a very superficial level, and some essential control groups seem to be missing (unless I have misunderstood some of the methods). Details are provided below.

Critique:

1. The paper needs a solid edit from a native English speaker for language, grammar, spelling, and usage.
2. Details of the human population need to be provided, including exclusion criteria, status about the women's menstrual cycles (or use of steroid contraceptives), time of day of collection, and other experimental details.
3. Were field blanks used for the measures of BHPF in human serum? This is essential and appears to have been omitted.
4. In the yeast assay, the range of dosages used for BHPF should be extended into the lower range. EDCs are well-established as acting with non-monotonic dose-response curves and sometimes low dose effects are seen in the absence of high dose effects.
5. In vivo work on mice evaluates crude gross morphological changes that are crude that endocrinologists agree are poor measures of either estrogenic or anti-estrogenic activity. Neither the established uterotrophic assay is, or the author's new anti-uterotrophic assay, is a cutting-edge assay of hormonal actions. Similarly, global gene expression profiling of whole organs is not usually terribly informative, as tissues are highly heterogeneous.
6. By "intra-gastric administration" do the authors mean oral gavage, or were they surgically implanted with a feeding tube? This requires clarification. Moreover, gavage is highly stressful and should be avoided.
7. Justification and clarification of the dose of BHPF needs to be provided. In the anti-uterotrophic assay, the authors say administration was 5 mL/kg BW which is not meaningful. For the qPCR work, authors mention 50 mg/kg BW. If this was the dose given in prior studies, it is an unrealistic dose and not relevant to human exposures. Subsequent work on subchronic toxicity uses a range of dosing from 0.4 to 50 mg/kg, still in a high range.

8. The authors might consider doing the obvious experiment of allowing the mice to drink from water bottles with BHPF, compared to a vehicle water bottle. This would avoid the gavage problem and would use realistic amounts.
9. Mating studies were conducted with same-treatment males and females. It is important to mate treated animals with non-treated controls.
10. In Figure 4 legend, clarify that heatmaps are shown relative to the control group.
11. The discussion of NOAEL needs to include work on much lower dosages, showing adverse effects well below the predicted NOAEL.
12. There are other mechanisms of action of bisphenols beyond estrogen signaling that should be discussed.

Reviewer #3 (Remarks to the Author):

General Comments:

First, I note that the Editor asked me to pay specific attention to the analytical-chemistry and the statistical aspects of this work. These are certainly most central to my expertise; however, I am also reasonable well qualified to judge the (anti-)estrogenic assays and microarray results. The histopathology and some details of the in vivo studies are the only aspects of the paper that are beyond my comfortable reach.

This is a very thorough, compelling, and interesting piece of work. The Authors conducted experiments that span a wide range of disciplines, assembling all the pieces needed to strongly "make their case". Specifically, they show conclusively that: 1) fluorene-9-bisphenol (BHPF) occurs in plastic bottles (using ¹H-NMR and ¹³C-NMR), 2) BHPF leaches from bottles into drinking water (using GC-MS with an in-house synthesized deuterated form of BHPF as an internal standard), 3) BHPF is detected (albeit at low occurrence rate) in serum in the general public (using 100 volunteers), and 4) BHPF is a potent anti-estrogen. To establish the anti-estrogenic mode of action, the Authors use multiple lines of persuasive evidence demonstrating that BHPF: a) blocks the activity of estradiol (on par with a model anti-estrogen) in a well-established yeast assay for estrogenicity, b) fits nicely within the antagonist pocket (but not the agonist pocket) of the estrogen receptor using in silico molecular docking software, c) inhibits relative uterine weight (in a dose-dependent manner) in an optimized in vivo screen for anti-estrogenicity (in similar fashion with a model anti-estrogen), d) selectively down-regulates (in a dose-dependent manner) virtually all of the transcripts that are up-regulated by estradiol in mouse microarrays, e) reduces relatively uterine weight in reproductive

toxicology in vivo studies (as do model anti-estrogens), and f) impacts tissues similar to model anti-estrogens as viewed by histology.

All of these lines of evidence are laid out in a very logical fashion. The Authors use these results to illustrate the larger implications that: a) alternatives to BPA may be as harmful, or more harmful, to the public as BPA, b) thus, the various pressures (regulations, public concerns, etc.) that force companies to replace "hot button" chemicals may be doing more harm than good, c) the topic of anti-estrogens in the environment - much less studied than that of estrogens - deserves more attention from researchers. I have heard each of these larger points a few times before, although each is "fresh" enough that I would consider them relatively novel. Indeed, I cannot recall a single case where these points were made in a more-compelling and thorough manner. I feel strongly that this paper would be of interest to those in the Environmental Science community. On a personal note, I am involved in work that uses chemical monitoring data from waste water treatment plants. Of course, I will abide by the Journal's rules of confidentiality, but I am very curious to know if BHPF could be measured in any of these samples. I do feel that this paper, once published, will likewise impact the thinking of others in this field.

My recommendation is to accept this paper with minor modifications. I have read the scope and criteria for publication for Nature Communications, and I think this paper is well suited. Following are some specific comments that are aimed at improving the presentation (and also some comments regarding the Editor's specific charge to assess the statistical and analytical methodology).

1) With regard to statistics - there really is nothing "fancy" here - which is as it should be. The Authors use 10 replicates, which is about as good as you see with in vivo rodent studies. A well-established statistical program is used; differences in classes are assessed with ANOVA using Fisher's as a post-hoc test. In the figures, error bars are generated using standard deviation, which, to their credit, is more conservative than standard-error-of-the-mean error bars (which is often used). If I wanted to be nitpicky, I would point out that Fisher's test is a little less conservative than Tukey's test, but that is really more of a personal preference, and would probably not make any difference. Also, the Authors do not state whether or not they tested the data for normality and heteroscedasticity before applying the ANOVA tests. Strictly speaking, ANOVA is only applicable for data that meet these criteria (although a small degree of non-adherence is well tolerated). Looking at the data, I suspect there are no problems, but it might be worthwhile to close that loop.

2) With regard to analytical measurements - The Author's choices for analyses seem perfectly appropriate. They initially used ¹H-NMR and ¹³C-NMR to confirm that BHPF was leaching from a plastic bottle. BHPF was isolated by fractionating a methanol leachate. NMR is the "gold standard" for identifying (or confirming the identity) of a relatively pure organic chemical. Using both ¹H and ¹³C NMR takes this analysis to a very high level of confidence. My only quibble with this part of the work is that I would have liked to see a Figure with a more explicit comparison of the NMR spectrum

of the isolate with that of an authentic standard of BHPF. I feel sure that the Authors have made this comparison for their own sake - indeed, I think the NMR spectrum in Supplementary Information Figure 1A is for the authentic standard. However, this is not clearly stated, and it is not presented in a way that can be compared directly to the NMR spectrum of the isolate in Figure 1.

Once identified, the Authors used GC/MS (with derivatization) to quantify BHPF in drinking water and in human serum. They included a partially deuterated form of BHPF (which was synthesized in-house) as an internal standard. Again, this is the "gold standard" method for target analysis (and quantification) of an organic chemical whose mass spectrum and GC retention time is known. No qualms here.

3) It seems that there has been some inversions (typos) when referring to Figures and sub-parts of Figures. For example, in the Caption, Figure 7D is described as "dead fetus ...". It seems pretty clear that Figure 7C is actually the dead fetus (and it is referred to as such in the text). In addition, the "order" of Figure 5 and Figure 6 (the actual graphics) are switched (6 appears in the document before 5). And, it appears that some text references to 5/6 are reversed. I am not sure I caught all of these issues, so beware!

4) The microarray results regarding the opposing effects of BHPF and E2 (described from lines 154 - 170, and depicted in Figure 4) are very compelling and useful to the argument. However, the more "global" microarray-results discussion in lines 127 - 154 is not very useful. While there are a few good points in that section, in my view, the vast majority of the text from line 127 - 154 could be omitted or perhaps moved to Supplementary Information.

5) I thought the paper would be more impactful if the Discussion had ended with line 283, which is an effective climactic sentence. The paragraph that follows (line 284-293) is mostly a repetitive summary of the technical findings. I suggest moving any useful thoughts from lines 284-293 to earlier in the Discussion, and closing with the preceding paragraph that ends on line 283.

6) The English linguistics of this paper is quite good; however, some issues will need to be addressed. For example, "drinking water" is called "drink water". I will not list more here, but several other subtle misusages appear.

Finally, I would note that the references seemed appropriate, and that all sections of the manuscript were clear, lucid, and very well written and organized.

Reviewer #4 (Remarks to the Author):

Molecular docking of BHPF

Molecular docking was performed using the commercial software Scigress (Ultra Version 3.0.0, Fujitsu). It shows that BHPF can be accommodated into the antagonist pocket of ER α (PDB ID 3ERT), and not in the agonist pocket (PDB ID 1ERE)(Fig. 2c-e). The position of BHPF in the pocket is rather similar to that of OHT observed in the crystal structure. The approach used is a standard one. Considering the stereochemistry of the two ligands the results are convincing and not surprising.

A comparative analysis with the crystal structures of ER in complex with BPA and BPC (ref 32, Delfosse et al. Proc. Natl. Acad. Sci. USA 109, 14930-14935 (2012)) would be useful.

Response to Reviewer #1's Comments:

Comment A. Summary. This manuscript describes work showing that BHPF is an antiestrogen, that it can leach from some plastic water bottles, that it can be found in human serum. These are important and original observations that have important implications globally.

Response: Thank you.

Comment B. This is highly original and of great interest.

Response: Thank you.

Comment C. Data and methodology. This reviewer cannot evaluate the quality of the chemistry approaches. However, there are potential concerns for consideration. First, the YES assay has certain weaknesses associated with the mosaic nature of the target receptor. It would strengthen the body of work (which is considerable already) to use a full length human receptor and evaluate separate effects on ERa and ERb. Recognizing that this may be deemed to be outside the scope of this original paper. The description of the data in humans should be perhaps more completely described. First, this should not be described as "general population", but rather volunteers. How were these volunteers identified and selected? What age, sex, etc. Other characteristics? This self-selection process could produce a bias. Second, are the authors confident that the measurements of BHPF in human serum is not the result of contamination from plastics employed in the procedure? If no field blanks or tests of equipment was performed, the authors need to be very careful about these conclusions. Moreover, there were very few volunteers that exhibited BHPF levels and this should be reflected in the abstract and summaries ("...was detected in a small proportion of volunteers"...). The animal studies appear appropriate. However, it might be useful to describe the methods employed to reduce the risk of bias in data collection. Method of animal assignment, sequence of gavage and animal collection, time of day, etc. A separate subsection in the supplementary material might be useful for this.

Response: Thank you very much for your very helpful and constructive comments. To specifically answer these comments, we have divided the “Comment C” into 3 parts and responded the comments point by point, as follows:

Comment C-1. Data and methodology. This reviewer cannot evaluate the quality of the chemistry approaches. However, there are potential concerns for consideration. First, the YES assay has certain weaknesses associated with the mosaic nature of the target receptor. It would strengthen the body of work (which is considerable already) to use a full length human receptor and evaluate separate effects on ER α and ER β . Recognizing that this may be deemed to be outside the scope of this original paper.

Response: According to your suggestion, we performed dual-luciferase reporter assays which comprised a full-length human estrogen receptor α or β , and the anti-estrogenicity of BHPF have been demonstrated by the DLR assays. The following sentences and Fig. 2c–2f were added in the revised manuscript.

“The dual-luciferase reporter assay comprised a full-length human estrogen receptor α or β and estrogen response elements (EREs).” (page 5, lines 92-93)

“Similarly, BHPF showed no estrogenic activity but strong anti-estrogenic activities in the dual-luciferase reporter assays (Fig. 2c–2f). The IC₅₀ values of BHPF were 1.09×10^{-7} and 7.53×10^{-8} M for estrogen receptor α and estrogen receptor β , respectively, when BHPF coexisted with 1×10^{-9} M E₂. A WST-1 cell proliferation assay using human choriocarcinoma JEG-3 cells was performed, and no obvious cytotoxicity of BHPF was observed from 1.0×10^{-6} M to lower concentrations (Supplementary Fig. 2). The IC₅₀ values of OHT were 8.61×10^{-9} and 5.19×10^{-9} M for estrogen receptor α and estrogen receptor β , respectively, which were 12.66 and 14.51 times lower than those of BHTF in the dual-luciferase DLR assays.” (page 5, lines 99-106)

“Dual-luciferase reporter assays. *Dual-luciferase reporter assays were carried out primarily according to the method described in a previous study³⁹. In short,*

*full-length sequences of human estrogen receptor α and β were amplified by RT-PCR using mRNA from MCF-7, and inserted into a pSVSPORT1 (Invitrogen); the resulting constructs were termed pSVhER α and pSVhER β . A reporter plasmid containing ERE was constructed by inserting four tandem repeats of an ERE of *Xenopus laevis* vitellogenin A2 into a pGL4.23 vector (Promega), and was termed pGL4-ERE-luc. All of the plasmids constructed were confirmed by sequence analysis. The Renilla LUC control reporter construct pGL4.74-TK was purchased from Promega (Madison, WI). Human choriocarcinoma JEG-3 cells (ATCC No. HTB-36) were obtained from the American Type Culture Collection (ATCC; Manassas, VA) and were cultured in minimal essential medium (MEM, Invitrogen, Carlsbad, CA) containing 2 mM L-glutamine, 0.1 mM minimal essential medium nonessential amino acid solution (Invitrogen/Thermo Fisher Scientific, Grand Island, NY), and 10% fetal calf serum (FCS) at 37°C in a humidified atmosphere containing 5% CO₂. The cells (3×10^4 cells/well) were seeded in 24-well plates 24 h before transfection. The cells were transfected with pGL4-ERE-luc (5 ng/well), pGL 4.74-TK (0.2 ng/well) and either pSVhER α (5 ng/well) or pSVhER β (5 ng/well) using Lipofectamine reagent (Invitrogen). At 24 h after transfection, the compounds in DMSO were added to the cultures at a volume ratio (v/v) of 0.1%. The cells were continuously cultured in medium supplemented with 1% charcoal-stripped FCS for another 24 h and then harvested, and cell extracts were assayed for firefly luciferase activity. The assays were performed in quadruplicate, and the results were expressed as the fold induction of the control after normalization against Renilla luciferase activity. Sigmoidal dose-effect curves were calculated using the GraphPad Prism 4 software. The IC₅₀ values were calculated on the basis of the sigmoidal dose-effect curves. Finally, a WST-1 cell proliferation assay (see Supplementary Information) was performed to evaluate the*

cytotoxicity of BHPF using the cell proliferation reagent WST-1 (Dojindo, Mashiki, Japan).” (pages 21-22, lines 445-470)

Comment C-2. The description of the data in humans should be perhaps more completely described. First, this should not be described as "general population", but rather volunteers. How were these volunteers identified and selected? What age, sex, etc. Other characteristics? This self-selection process could produce a bias. Second, are the authors confidence that the measurements of BHPF in human serum is not the result of contamination from plastics employed in the procedure? If no field blanks or tests of equipment was performed, the authors need to be very careful about these conclusions. Moreover, there were very few volunteers that exhibited BHPF levels and this should be reflected in the abstract and summaries ("...was detected in a small proportion of volunteers"...).

Response: According to your suggestion, we replaced the "general population" with "volunteers", added "be detectable in the serum of a small proportion of human volunteers" in the abstract and summaries, and added more clinical information and experimental details on the human volunteers and serum sample measurements in the revised manuscript. The following sentences were added in the revised manuscript:

"...who habitually used plastic bottles for drinking water." (page 4, line 86)

"In China, college students usually undergo a physical examination before graduation. The volunteers in this study were senior students who were undergoing a physical examination in Beijing. Fifty male and 50 female healthy volunteers (mean age, 23.5 ± 1.2 years) who habitually used plastic water bottles were randomly recruited to participate in this study in June 2015..." (page 19, lines 406-410)

"Fasting blood was collected at Peking University Hospital Medical Center with vacutainer blood collection device (Chengwu Yongkang Medical Products Ltd., China) between 8 and 9 AM. The blood collection device principally consisted of a

steel needle with sharp ends, which linked the blood-vessel lumen with the glass vacutainer tube during blood collection, and a polypropylene auxiliary syringe for holding the blood, which had been verified to be free of BHPF contamination.” (pages 19-20, lines 413-418)

“All analytical procedures were checked for precision, reproducibility, blank contamination, and linearity. Quality control was maintained by analyzing a method blank (calf serum) and two spiked calf serum samples (spiked with BHPF-d₇ singly or a mixture of BHPF-d₇ and undeuterated BHPF) along with every 12 samples. No BHPF was detected in the blank and single BHPF-d₇ spiked calf serum samples during the analytical procedures.” (page 20, lines 429-434)

“In this study, we recruited 100 college student volunteers (50 female and 50 male) who habitually drank from plastic bottles, and measured their serum BHPF levels by GC-MS. It was found that BHPF was detectable in 7 volunteers (4 male and 3 female), with a mean serum BHPF level of 0.34 ± 0.21 ng/mL and a maximal level of 0.7 ng/mL. In female CD-1 mice given cooled boiled water with a concentration of 124.35 ng/L of BHPF released from plastic bottles, ad libitum for 10 days beginning on PND 24, serum BHPF levels of 1.21 ± 1.11 ng/mL were detected in the group receiving water from Bottle A, suggesting that drinking cooled boiled water from plastic bottles may be sufficient to elevate the BHPF levels in human blood. In China and some other countries, drinking boiled water is routine. Plastic bottles, being easily transportable, are often used to store boiled drinking water. Student populations (from elementary students to college students) are among the most common users of plastic bottles. Many students carry plastic bottles every day, and mostly drink boiled water collected from public drinking-water heaters. However, because students (particularly elementary and middle school students) are at

developmental stages from prepuberty to sexual maturity, the effects of anti-estrogenic pollutants on these populations deserve particular attention.” (pages 14-15, lines 301-316)

Comment C-3. The animal studies appear appropriate. However, it might be useful to describe the methods employed to reduce the risk of bias in data collection. Method of animal assignment, sequence of gavage and animal collection, time of day, etc. A separate subsection in the supplementary material might be useful for this.

Response: According to your suggestion, we added “Supplementary description of animal experiments” in the Supplementary Information:

“Supplementary description of animal experiments. Immature female CD-1 mice were obtained from Experimental Animal Tech Co. of Weitonglihua (Beijing, China). Mice with a maximal difference in body weight of 1 g were selected for the experiments and randomly assigned to either the treatment or control groups. For the oral gavage treatment experiments, the mice in each cage were labeled by shaving hair on different parts of the body, and the oral gavage treatment was performed in turns for each group, whereby in each turn only one mouse in a group was treated. After the period of treatment, the mice were weighed and sacrificed 24 h after the final treatment according to the sequence of the treatments, and the uteri were removed, blotted, and weighed by one experimenter to reduce the risk of bias in the data collection. For the drinking water treatment experiments, the mice were weighed and sacrificed in turns for each group, whereby in each turn only one mouse in a group was sacrificed; the uteri were removed, blotted, and weighed by one experimenter to reduce the risk of bias in the data collection.”
(Supplementary Information)

Comment D. Statistics and Uncertainties. The human data may be the most uncertain and this should be clearly stated.

Response: According to your suggestion, we added more clinical information and experimental details on the human volunteers and serum sample measurements in the revised manuscript. The following sentences were added in the revised manuscript:

“Fasting blood was collected at Peking University Hospital Medical Center with vacutainer blood collection device (Chengwu Yongkang Medical Products Ltd., China) between 8 and 9 AM. The blood collection device principally consisted of a steel needle with sharp ends, which linked the blood-vessel lumen with the glass vacutainer tube during blood collection, and a polypropylene auxiliary syringe for holding the blood, which had been verified to be free of BHPF contamination.” (pages 19-20, lines 413-418)

“All analytical procedures were checked for precision, reproducibility, blank contamination, and linearity. Quality control was maintained by analyzing a method blank (calf serum) and two spiked calf serum samples (spiked with BHPF-d₇ singly or a mixture of BHPF-d₇ and undeuterated BHPF) along with every 12 samples. No BHPF was detected in the blank and single BHPF-d₇ spiked calf serum samples during the analytical procedures.” (page 20, lines 429-434)

Comment E. Conclusions are appropriate. However, it may be important to point out that the US EPA does not employ an "antagonist mode" in their estrogen receptor HTS assays in ToxCast because environmental anti-estrogens have not been identified. The work here has very important implications for this.

Response: In a review article previously published by Rotroff et al. (2013), it was found that “estrogen receptor- α -antagonist” assay was listed in HTS assays of ToxCast, so it is not sure that the US EPA have not employed an "antagonist mode" in

their estrogen receptor HTS assays in ToxCast so far. But we think that your suggestion is very constructive, and according to the suggestion, we added the following sentences in the conclusions of the revised manuscript:

“Screening programs for endocrine disruptors have been established in many countries in recent years, but many of these programs have neglected to screen anti-estrogens. The results of this study suggest that anti-estrogenic pollutants, as well as their adverse effects on human health, should be of concern in the future.”

(page 17, lines 358-361)

Table 1. Summary of endocrine-related HTS assays.

ToxCast assay	Assigned MOA	Species	Assay target	Assay technology	Chemicals tested (n)		
					Unique	Overlapping with EDSP/OECD	Overlapping with active chemicals in ToxCast
ATG_AR_TRANS	HTS-A	Human	Androgen receptor-agonist	Multiplexed reporter gene assay	309 ^a	13	0
NCGC_AR_Agonist	HTS-A	Human	Androgen receptor-agonist	GAL4 BLAM reporter gene assay	309	13	0
NCGC_AR_Antagonist	HTS-A	Human	Androgen receptor-antagonist	GAL4 BLAM reporter gene assay	309	13	5
NVS_NR_hAR	HTS-A	Human	Androgen receptor	Competitive binding	309	13	6
NVS_NR_rAR	HTS-A	Rat	Androgen receptor	Competitive binding	309	13	1
ATG_ERa_TRANS	HTS-E	Human	Estrogen receptor- α	Multiplexed reporter gene assay	326 ^b	21	12
ATG_ERE_CIS	HTS-E	Human	Estrogen receptor response element	Multiplexed reporter gene assay	326 ^b	21	11
ATG_ERRa_TRANS	HTS-E	Human	Estrogen related receptor- α	Multiplexed reporter gene assay	326 ^b	21	0
ATG_ERRg_TRANS	HTS-E	Human	Estrogen related receptor- γ	Multiplexed reporter gene assay	326 ^b	21	0
NCGC_ERalpha_Agonist	HTS-E	Human	Estrogen receptor- α -agonist	GAL4 BLAM reporter gene assay	326 ^b	21	7
NCGC_ERalpha_Antagonist	HTS-E	Human	Estrogen receptor- α -antagonist	GAL4 BLAM reporter gene assay	309	15	4
NVS_NR_bER	HTS-E	Bovine	Estrogen receptor	Competitive binding	316 ^b	17	1
NVS_NR_hER	HTS-E	Human	Estrogen receptor	Competitive binding	326 ^b	21	4
NVS_NR_mERa	HTS-E	Mouse	Estrogen receptor- α	Competitive binding	316 ^b	17	1
NVS_ADME_hCYP19A1	HTS-S	Human	Aromatase	Enzyme Inhibition	309	17	1
NCGC_TRbeta_Agonist	HTS-T	Human	Thyroid hormone receptor- β -agonist	GAL4 BLAM reporter gene assay	309	8	0
NCGC_TRbeta_Antagonist	HTS-T	Human	Thyroid hormone receptor- β -antagonist	GAL4 BLAM reporter gene assay	309	8	0
NVS_NR_hTRa	HTS-T	Human	Thyroid hormone receptor- α -antagonist	Receptor activation	309	8	0

^aAdditional reference compounds from Judson et al. (2010) were run but not included because this is the only androgen-related HTS assay that tested these chemicals. ^bIncludes additional reference compounds from Judson et al. (2010).

References

Rotroff, D. M. et al. Using in vitro high throughput screening assays to identify potential endocrine-disrupting chemicals. *Environ Health Perspect.* **121**, 7-14, (2013).

Comments F-H.

F. Improvements listed above.

G. References are appropriate

H. Well-written manuscript.

Response: Thanks.

Response to Reviewer #2's Comments:

General Comments. This study identified a BPA substitute, BHPF, in plastic water bottles, found that it is detectable in the water through leaching at 60C, and tested its potential actions as an endocrine-disrupting chemicals (EDC), particularly as an anti-estrogen through in vitro and in vivo experiments. They also showed that BHPF is detectable in humans. The identification of EDC activity of replacement chemicals is very important, and the study implicates BHPF as another such chemical, alongside other bisphenols such as BPS, BPF, and others. The major strength of work is making comparisons across all of these different levels. There are also novel aspects of work such as the docking experiment. The weakness is that much of the work is done at a very superficial level, and some essential control groups seem to be missing (unless I have misunderstood some of the methods). Details are provided below.

Response: Thank you very much for your comments and kind suggestions. To increase technical nature of the work, we added dual-luciferase reporter assays which comprised a full-length human estrogen receptor α or β . Dual-luciferase reporter assay is a relatively new research tool. In this study, each experiment has a negative (vehicle) control group and a positive control group. According your comment, we revised the manuscript to make the manuscript more legible.

Critique:

Comment 1. The paper needs a solid edit from a native English speaker for language, grammar, spelling, and usage.

Response: According to your comment, two native English language editors had help us to revise the manuscript.

Comment 2. Details of the human population need to be provided, including exclusion criteria, status about the women's menstrual cycles (or use of steroid contraceptives), time of day of collection, and other experimental details.

Response: According to your suggestions, we added more clinical information and experimental details on the human volunteers and serum sample measurements in the revised manuscript. The following sentences were added in the revised manuscript:

“In China, college students usually undergo a physical examination before graduation. The volunteers in this study were senior students who were undergoing a physical examination in Beijing. Fifty male and 50 female healthy volunteers (mean age, 23.5 ± 1.2 years) who habitually used plastic water bottles were randomly recruited to participate in this study in June 2015...” (page 19, lines 406-410)

“Fasting blood was collected at Peking University Hospital Medical Center with vacutainer blood collection device (Chengwu Yongkang Medical Products Ltd., China) between 8 and 9 AM. The blood collection device principally consisted of a steel needle with sharp ends, which linked the blood-vessel lumen with the glass vacutainer tube during blood collection, and a polypropylene auxiliary syringe for holding the blood, which had been verified to be free of BHPF contamination.” (pages 19-20, lines 413-418)

“All analytical procedures were checked for precision, reproducibility, blank contamination, and linearity. Quality control was maintained by analyzing a method blank (calf serum) and two spiked calf serum samples (spiked with BHPF-d₇ singly or a mixture of BHPF-d₇ and undeuterated BHPF) along with every 12 samples. No BHPF was detected in the blank and single BHPF-d₇ spiked calf serum samples during the analytical procedures.” (page 20, lines 429-434)

Comment 3. Were field blanks used for the measures of BHPF in human serum? This is essential and appears to have been omitted.

Response: Thank you for your comment. Quality control was maintained by analyzing a method blank (calf serum) and two spiked calf serum samples along with every 12 samples. No BHPF was detected in the blank and spiked calf serum samples

during the analytical procedures. The blood collection device had been verified to be free of BHPF contamination when used for measurement of serum BHPF in mice. In addition, BHPF was not detected in most serum samples in human volunteers suggested that there is no BHPF contamination during blood collection. According to your suggestions, we added more experimental details on BHPF measurements in the revised manuscript. The following sentences were added in the revised manuscript:

“Fasting blood was collected at Peking University Hospital Medical Center with vacutainer blood collection device (Chengwu Yongkang Medical Products Ltd., China) between 8 and 9 AM. The blood collection device principally consisted of a steel needle with sharp ends, which linked the blood-vessel lumen with the glass vacutainer tube during blood collection, and a polypropylene auxiliary syringe for holding the blood, which had been verified to be free of BHPF contamination.” (pages 19-20, lines 413-418)

“All analytical procedures were checked for precision, reproducibility, blank contamination, and linearity. Quality control was maintained by analyzing a method blank (calf serum) and two spiked calf serum samples (spiked with BHPF-d₇ singly or a mixture of BHPF-d₇ and undeuterated BHPF) along with every 12 samples. No BHPF was detected in the blank and single BHPF-d₇ spiked calf serum samples during the analytical procedures.” (page 20, lines 429-434)

Comment 4. In the yeast assay, the range of dosages used for BHPF should be extended into the lower range. EDCs are well-established as acting with non-monotonic dose-response curves and sometimes low dose effects are seen in the absence of high dose effects.

Response: Thank you for your comment. In the yeast assay, we had studied lower concentrations of BHPF in preliminary experiments, and no obvious effect was found.

In the dual-luciferase reporter assays, we studied lower concentrations of BHPF and found BHPF took anti-estrogenic effect at lower concentrations. According to your suggestions, we added the results of dual-luciferase reporter assays in Fig. 2 in the revised manuscript, and the following sentences were added in the revised manuscript.

“Similarly, BHPF showed no estrogenic activity but strong anti-estrogenic activities in the dual-luciferase reporter assays (Fig. 2c–2f). The IC_{50} values of BHPF were 1.09×10^{-7} and 7.53×10^{-8} M for estrogen receptor α and estrogen receptor β , respectively, when BHPF coexisted with 1×10^{-9} M E_2 .” (page 5, lines 99-102)

Comment 5. In vivo work on mice evaluates crude gross morphological changes that are crude that endocrinologists agree are poor measures of either estrogenic or anti-estrogenic activity. Neither the established uterotrophic assay is, or the author's new anti-uterotrophic assay, is a cutting-edge assay of hormonal actions. Similarly, global gene expression profiling of whole organs is not usually terribly informative, as tissues are highly heterogeneous.

Response: Thank you for your comments. As you commented, in vivo work using mice is difficult to use for evaluating the estrogenic or anti-estrogenic activity of chemical sometimes. Many factors, such as body weight, age and strain of mice and experimental procedure, may affect the results during experiment. Fortunately, in this study, some of the authors have focally studied on animal experiments for decades and have rich experiences with the study of mice. The selections of body weight, age and strain of mice, as well as the experimental procedure, were resulted from a lot of preliminary experiments. We agree with you that the uterotrophic assay is not a cutting-edge assay of hormonal actions. But the uterotrophic assay is valuable in evaluating strong estrogenic or anti-estrogenic compounds and BHPF is a strong anti-estrogenic compound. After all, the uterotrophic assay is still the most common

used in vivo tool for evaluating estrogenic or anti-estrogenic activity of chemicals and is adopted by OECD (OECD Test No. 440), US EPA (OCSPP Guideline 890.1600), etc. As you commented, global gene expression profiling of whole organs is not usually terribly informative because tissues are highly heterogeneous. In this study, the total RNA from each pool of tissue samples was a mixture from three animals; it is helpful to reduce the bother of heterogeneity.

“The total RNA from each pool of tissue samples (a mixture from three animals) was used for microarray analysis.” (page 24, lines 519-520)

References

OECD Test No. 440: Uterotrophic Bioassay in Rodents : A short-term screening test for oestrogenic properties.

<https://ntp.niehs.nih.gov/iccvam/suppdocs/feddocs/oecd/oecd440.pdf>

U.S. Environmental Protection Agency. 2011. Uterotrophic Assay. OCSPP Guideline 890.1600. Standard Evaluation Procedure (SEP). Endocrine Disruptor Screening Program.

https://www.epa.gov/sites/production/files/2015-07/documents/final_890.1600_uterotrophic_assay_sep_9.22.11.pdf

Comment 6. By "intra gastric administration" do the authors mean oral gavage, or were they surgically implanted with a feeding tube? This requires clarification. Moreover, gavage is highly stressful and should be avoided.

Response: According to your suggestions, we changed the “intra gastric administration” to “oral gavage”, and an exposure experiment through drinking water using mice was performed to avoid stress caused by gavage in the revised manuscript. We agree with you that gavage may induce stress. But oral gavage is still widely employed in animal experiments and accepted by standard evaluation procedures of OECD (OECD Test No. 440), US EPA (OCSPP Guideline 890.1600), etc. In addition, it should be pointed out that: 1) the operators for oral gavage are highly skilled in this study and highly skilled operators would greatly reduce the stress on mice; 2) the mice were pre-gavaged with vehicle before dosing for acclimatization; 3) the mice of control

group were treated by gavage in the same way as those of test groups. These measures helped to prevent influences of gavage on experiment.

References

OECD Test No. 440: Uterotrophic Bioassay in Rodents : A short-term screening test for oestrogenic properties.

<https://ntp.niehs.nih.gov/iccvam/suppdocs/feddocs/oced/ocedtg440.pdf>

U.S. Environmental Protection Agency. 2011. Uterotrophic Assay. OCSP Guideline 890.1600. Standard Evaluation Procedure (SEP). Endocrine Disruptor Screening Program.

https://www.epa.gov/sites/production/files/2015-07/documents/final_890.1600_uterotrophic_assay_sep_9.22.11.pdf

Comment 7 and 8. Justification and clarification of the dose of BHPF needs to be provided. In the anti-uterotrophic assay, the authors say administration was 5 mL/kg BW which is not meaningful. For the qPCR work, authors mention 50 mg/kg BW. If this was the dose given in prior studies, it is an unrealistic dose and not relevant to human exposures. Subsequent work on subchronic toxicity uses a range of dosing from 0.4 to 50 mg/kg, still in a high range. The authors might consider doing the obvious experiment of allowing the mice to drink from water bottles with BHPF, compared to a vehicle water bottle. This would avoid the gavage problem and would use realistic amounts.

Response: Thank you for your comments and suggestions. 1) The doses of BHPF for animal experiments were designed in reference to in vitro results at beginning, and then to other animal experiments previously performed. Meanwhile, in the experiments with longer exposure period, the lower doses were selected for further studies. 2) The “5 mL/kg BW” was the volume of vehicle or chemical solutions administered. According to your comment, the original sentence with “5 mL/kg BW” was revised to make it legible. 3) The experiment with 50 mg/kg BW group, as well as a control, was specifically conducted for the expression profiling and Q-RT-PCR works. According to your comment, we added more details about the experiment. 4) According to the suggestion of “experiment of allowing the mice to drink from water bottles with BHPF”, we performed exposure experiment through drinking water using

mice, and studied the effects of low doses of BHPF relevant to human exposure. The following sentences were added or revised in the revised manuscript:

“The volume of vehicle or chemical solutions administered was adjusted daily based on body weight measured during the dosing period according to the volume–body weight ratio of 5 mL/kg bw.” (page 23, lines 495-498; pages 25-26, lines 548-550)

“Immature female CD-1 mice were obtained from Experimental Animal Tech Co. of Weitonglihua (Beijing, China) and acclimatized in an experimental environment with a temperature of 22°C ± 2°C, relative humidity between 40% and 60%, and artificial lighting in a 12 h/12 h light–dark cycle. The animals were fed ad libitum with a basic diet from the Laboratory Animal Center of the Academy of Military Medical Sciences (Beijing, China), and drinking water was provided ad libitum. The mice were treated with 50 mg/kg bw of BHPF or peanut oil via oral gavage for 3 days beginning on PND 21. On PND 24, mice of each group were weighed and sacrificed by cervical dislocation. The uteri, ovaries, and livers were collected for total RNA isolation. Total RNA was isolated using Trizol reagent (Invitrogen) and further purified by NucleoSpin RNA Clean-up (Macherey-Nagel, Germany).” (page 24, lines 507-517)

*“**Effects of low doses of BHPF relevant to human exposure.** To study the effects of BHPF at doses relevant to human exposure, water samples with BHPF artificially added (100 ng/L BHPF) or released from “BPA-free” plastic water bottles (Bottle A and Bottle B) were given ad libitum to female CD-1 mice beginning on PND 24. The mice of the Bottle A and Bottle B groups received cooled boiled water that had been filled into Bottle A and Bottle B, respectively, while still boiling, and their BHPF levels were determined to be 124.35 and 23.81 ng/L, respectively. After a 10-day exposure, the relative uterine weights in the groups of 100 ng/L BHPF, Bottle A, Bottle B, and the positive control (100 ng/L FULV) were decreased to 79.34% ± 26.99%,*

76.62% ± 19.97%, 94.81% ± 34.45%, and 77.89% ± 37.50% that of the control ($P > 0.05$), respectively (Fig. 8a). The gene expressions of *sprr2a* and *sprr2b* in the uteri of the mice were also studied by Q-RT-PCR (Fig. 8b). The expressions of *sprr2a* were decreased in the groups of 100 ng/L FULV, 100 ng/L BHPF, and Bottle A, but no statistically significant difference was observed ($P > 0.05$). The gene expressions of *sprr2b* were decreased in all of the test groups, and were significantly lower ($P < 0.05$) than that of the control in the groups of 100 ng/L FULV and Bottle A. Finally, the serum levels of BHPF in each mouse were determined after enzymatic hydrolysis using β -glucuronidase/arylsulfatase. Serum BHPF was detected only in the mice of the Bottle A group (1.21 ± 1.11 ng/mL), with BHPF levels in the range of 0.36–2.70 ng/mL, but no serum BHPF was detected in the 100 ng/L BHPF group.” (page 11, lines 221-239)

“Exposure experiment through drinking water using mice. Immature female CD-1 mice were obtained from Experimental Animal Tech Co. of Weitonglihua (Beijing, China). The animals were housed four to a cage and acclimatized in a controlled environment with a temperature of $22^{\circ}\text{C} \pm 2^{\circ}\text{C}$, relative humidity between 40% and 60%, and artificial lighting in a 12 h/12 h light–dark cycle. The animals were fed ad libitum an estrogen-free diet from Trophic Animal Feed High-tech Co., Ltd. (Nantong, China), and drinking water was provided ad libitum in glass bottles. Before the experiments, the mice were randomly assigned to five groups ($n = 8$). Ultra-pure water was used as a control. FULV was dissolved in ultra-pure water to a concentration of 100 ng/L and used as a positive control. BHPF was dissolved in ultra-pure water to prepare a concentration of 100 ng/L, which is relevant to human exposure. Two plastic drinking bottles labeled “BPA-free”, of different brands, were purchased and denoted Bottle A and Bottle B. The plastic bottles were filled with

*boiling ultra-pure water from a stainless steel electric water heater and allowed to cool down to room temperature prior to the animal experiment. The BHPF levels in the cooled boiled waters from Bottle A, Bottle B, and the stainless steel electric water heater were determined by GC-MS. No BHPF was detected in the cooled boiled water from the stainless steel electric water heater. The cooled boiled waters from Bottle A and Bottle B were transferred to glass bottles during the experiment. The exposure experiment was performed for 10 days beginning on PND 24. After the period of exposure, the mice were weighed and sacrificed. Blood was collected by cardiac puncture soon after each animal was sacrificed, and the uteri were removed, blotted, weighed, and immediately frozen in liquid nitrogen for gene expression analysis. The relative uterine weight was calculated to evaluate the anti-uterotrophic activity. The serum was separated by centrifugation and frozen at -20°C for BHPF analysis. The serum BHPF levels were analyzed by GC-MS using the same method as that for human serum. The gene expressions of *sprr2a* and *sprr2b* in the uteri were determined by Q-RT-PCR.”(pages 26-28, lines 570-594)*

Comment 9. Mating studies were conducted with same-treatment males and females. It is important to mate treated animals with non-treated controls.

Response: Thank you for your comment. In this study, in cases where pairing was unsuccessful, females were re-mated with proven males of the same group, except that females in the 1.2 mg/kg TAM group were re-mated with proven males of the control group because all of the females were non-pregnant in the TAM-treated group. This procedure is in reference to the mating procedures of reproduction toxicity Tests of OECD Guideline for Testing of Chemicals (OECD Test No. 416, 421, and 422).

“In cases where pairing was unsuccessful, females were re-mated with proven males of the same group, except that females in the 1.2 mg/kg TAM group were re-mated with proven males of the control group.” (page 26, lines 554-557)

References

OECD Test No. 416: Two-Generation Reproduction Toxicity Study. page 5.
OECD Test No. 421: Reproduction/Developmental Toxicity Screening Test. page 6.
OECD Test No. 422: Combined Repeated Dose Toxicity Study with the Reproduction/Developmental Toxicity Screening Test. page 7.

Comment 10. In Figure 4 legend, clarify that heatmaps are shown relative to the control group.

Response: According to your suggestion, we added “the fold change indicates the relative expression in the treatment versus the control” in the figure legend. (page 32, lines 744-745; 748-749).

Comment 11. The discussion of NOAEL needs to include work on much lower dosages, showing adverse effects well below the predicted NOAEL.

Response: It is known that the NOAEL is the highest experimental level of a chemical on animal or human that is without adverse effect under certain exposure condition. In the subchronic and reproductive toxicity tests of this study, mice were given doses of 0.4, 2, 10, and 50 mg/kg bw/3-d BHPF. We think that the dose of 0.4 mg/kg bw/3-d is much lower than the NOAEL (50 or 5 mg/kg bw/d) reported for BPA, so we compared the value with the NOAELs of BPA in discussion.

Comment 12. There are other mechanisms of action of bisphenols beyond estrogen signaling that should be discussed.

Response: According to your suggestion, we added other mechanisms of action of bisphenols beyond estrogen signaling in the discussion.

“Moreover, it should be noted that BHPF might induce anti-estrogenic effects through mechanisms other than nuclear estrogen receptors, as reported for BPA and some other bisphenols, which can induce adverse effects through mechanisms including the estrogen membrane receptor, estrogen-related receptor gamma, pregnane X receptor, etc^{2,16,20-22}. The microarray analysis also showed that some genes involved in

biotransformation and estrogen metabolism were up-regulated in the uteri of the BHPF-treated mice (Fig. 4b), and the up-regulation of these genes may facilitate the in utero metabolism of estrogen, thereby suppressing the effect of estrogen on the uterus.” (pages 12-13, lines 259-266)

Response to Reviewer #3's Comments:

General Comments: First, I note that the Editor asked me to pay specific attention to the analytical-chemistry and the statistical aspects of this work. These are certainly most central to my expertise; however, I am also reasonable well qualified to judge the (anti-)estrogenic assays and microarray results. The histopathology and some details of the in vivo studies are the only aspects of the paper that are beyond my comfortable reach.

This is a very thorough, compelling, and interesting piece of work. The Authors conducted experiments that span a wide range of disciplines, assembling all the pieces needed to strongly "make their case". Specifically, they show conclusively that: 1) fluorene-9-bisphenol (BHPF) occurs in plastic bottles (using $^1\text{H-NMR}$ and $^{13}\text{C-NMR}$), 2) BHPF leaches from bottles into drinking water (using GC-MS with an in-house synthesized deuterated form of BHPF as an internal standard), 3) BHPF is detected (albeit at low occurrence rate) in serum in the general public (using 100 volunteers), and 4) BHPF is a potent anti-estrogen. To establish the anti-estrogenic mode of action, the Authors use multiple lines of persuasive evidence demonstrating that BHPF: a) blocks the activity of estradiol (on par with a model anti-estrogen) in a well-established yeast assay for estrogenicity, b) fits nicely within the antagonist pocket (but not the agonist pocket) of the estrogen receptor using in silico molecular docking software, c) inhibits relative uterine weight (in a dose-dependent manner) in an optimized in vivo screen for anti-estrogenicity (in similar fashion with a model anti-estrogen), d) selectively down-regulates (in a dose-dependent manner) virtually all of the transcripts that are up-regulated by estradiol in mouse microarrays, e) reduces relative uterine weight in reproductive toxicology in vivo studies (as do model anti-estrogens), and f) impacts tissues similar to model anti-estrogens as viewed by histology.

All of these lines of evidence are laid out in a very logical fashion. The Authors use these results to illustrate the larger implications that: a) alternatives to BPA may be as harmful, or more harmful, to the public as BPA, b) thus, the various pressures (regulations, public concerns, etc.) that force companies to replace "hot button" chemicals may be doing more harm than good, c) the topic of anti-estrogens in the environment - much less studied than that of estrogens -

deserves more attention from researchers. I have heard each of these larger points a few times before, although each is "fresh" enough that I would consider them relatively novel. Indeed, I cannot recall a single case where these points were made in a more-compelling and thorough manner. I feel strongly that this paper would be of interest to those in the Environmental Science community. On a personal note, I am involved in work that uses chemical monitoring data from waste water treatment plants. Of course, I will abide by the Journal's rules of confidentiality, but I am very curious to know if BHPF could be measured in any of these samples. I do feel that this paper, once published, will likewise impact the thinking of others in this field.

My recommendation is to accept this paper with minor modifications. I have read the scope and criteria for publication for Nature Communications, and I think this paper is well suited. Following are some specific comments that are aimed at improving the presentation (and also some comments regarding the Editor's specific charge to assess the statistical and analytical methodology).

Response: Thank you very much for your kind and positive comments. We are pleased to write the response to your request about if BHPF could be measured in waste water treatment plants. Although we have not performed measurement of BHPF in waste water, we studied the BHPF pollution in urban surface water in 8 cities of China and an informal water sample from Mississippi River in Minneapolis. We found BHPF could be ubiquitously detected in urban surface water in China. The highest level of BHPF was detected in a water sample collected from a lake in Beijing, which was reached 54.8 ng/L. The level of BHPF was about 20.4 ng/L in the water sample from Mississippi River. So, we think BHPF should be detectable in waste water.

Comment 1. With regard to statistics - there really is nothing "fancy" here - which is as it should be. The Authors use 10 replicates, which is about as good as you see with in vivo rodent studies. A well-established statistical program is used; differences in classes are assessed with ANOVA using Fisher's as a

post-hoc test. In the figures, error bars are generated using standard deviation, which, to their credit, is more conservative than standard-error-of-the-mean error bars (which is often used). If I wanted to be nitpicky, I would point out that Fisher's test is a little less conservative than Tukey's test, but that is really more of a personal preference, and would probably not make any difference. Also, the Authors do not state whether or not they tested the data for normality and heteroscedasticity before applying the ANOVA tests. Strictly speaking, ANOVA is only applicable for data that meet these criteria (although a small degree of non-adherence is well tolerated). Looking at the data, I suspect there are no problems, but it might be worthwhile to close that loop.

Response: To test the normality and heteroscedasticity of the data before applying the ANOVA tests, we had performed One-Sample Kolmogorov-Smirnov Test and Test of Homogeneity of Variances prior to running one-way analysis of variance and Fisher's least significant difference tests. The data is normally distributed and the statistical methods used in this study should be appropriate. According to your suggestion, we added the following words "after running the one-sample Kolmogorov-Smirnov test and the test of homogeneity of variances" in the revised manuscript. (page 28, lines 598-599)

Comment 2. With regard to analytical measurements - The Author's choices for analyses seem perfectly appropriate. They initially used ¹H-NMR and ¹³C-NMR to confirm that BHPF was leaching from a plastic bottle. BHPF was isolated by fractionating a methanol leachate. NMR is the "gold standard" for identifying (or confirming the identity) of a relatively pure organic chemical. Using both ¹H and ¹³C NMR takes this analysis to a very high level of confidence. My only quibble with this part of the work is that I would have liked to see a Figure with a more explicit comparison of the NMR spectrum of the isolate with that of an authentic standard of BHPF. I feel sure that the Authors have made this comparison for their own sake - indeed, I think the NMR spectrum in Supplementary Information Figure 1A is for the authentic standard. However, this is not clearly stated, and it is not presented in a way that can be

compared directly to the NMR spectrum of the isolate in Figure 1. Once identified, the Authors used GC/MS (with derivatization) to quantify BHPF in drinking water and in human serum. They included a partially deuterated form of BHPF (which was synthesized in-house) as an internal standard. Again, this is the "gold standard" method for target analysis (and quantification) of an organic chemical whose mass spectrum and GC retention time is known. No qualms here.

Response: Thank you for your comments and suggestions. Because the NMR spectrum was reported previously by Liu et al. (2008), the NMR spectrum of BHPF standard was not shown in this manuscript, but we added the following sentence in the revised manuscript to make it legible according to your suggestion.

“The NMR results were consistent with those for BHPF reported previously¹¹.” (page 4, lines 69-70)

Comment 3. It seems that there has been some inversions (typos) when referring to Figures and sub-parts of Figures. For example, in the Caption, Figure 7D is described as "dead fetus ...". It seems pretty clear that Figure 7C is actually the dead fetus (and it is referred to as such in the text). In addition, the "order" of Figure 5 and Figure 6 (the actual graphics) are switched (6 appears in the document before 5). And, it appears that some text references to 5/6 are reversed. I am not sure I caught all of these issues, so beware!

Response: Thank you for your careful works on review this manuscript. According to your suggestions, we conducted a careful modification of the manuscript, and the errors were corrected.

Comment 4. The microarray results regarding the opposing effects of BHPF and E2 (described from lines 154 - 170, and depicted in Figure 4) are very compelling and useful to the argument. However, the more "global" microarray-results discussion in lines 127 - 154 is not very useful. While there are a few good points in that section, in my view, the vast majority of the text

from line 127 - 154 could be omitted or perhaps moved to Supplementary Information.

Response: According to your suggestion, these sentences (lines 127–154 in the original manuscript) were moved to Supplementary Information.

Comment 5. I thought the paper would be more impactful if the Discussion had ended with line 283, which is an effective climactic sentence. The paragraph that follows (line 284-293) is mostly a repetitive summary of the technical findings. I suggest moving any useful thoughts from lines 284-293 to earlier in the Discussion, and closing with the preceding paragraph that ends on line 283.

Response: According to your suggestion, the sentence was moved to the end of Discussion.

“Moreover, this study raises questions about the safety of BPA substitutes, and indicates the defect of the current chemical management for the substitution of hazardous chemicals.” (page 17, lines 361-363)

Comment 6. The English linguistics of this paper is quite good; however, some issues will need to be addressed. For example, "drinking water" is called "drink water". I will not list more here, but several other subtle misusages appear. Finally, I would note that the references seemed appropriate, and that all sections of the manuscript were clear, lucid, and very well written and organized.

Response: According to your suggestion, we conducted a careful modification of the manuscript and corrected the errors, and two native English language editors had help us to revise the manuscript.

Response to Reviewer #4's Comments:

Comments: Molecular docking was performed using the commercial software Scigress (Ultra Version 3.0.0, Fujitsu. It shows that BHPF can be accommodated into the antagonist pocket of ER α (PDB ID 3ERT), and not in the agonist pocket (PDB ID 1ERE)(Fig. 2c-e). The position of BHPF in the pocket is rather similar to that of OHT observed in the crystal structure. The approach used is a standard one. Considering the stereochemistry of the two ligands the results are convincing and not surprising. A comparative analysis with the crystal structures of ER in complex with BPA and BPC (ref 32, Delfosse et al. Proc. Natl. Acad. Sci. USA 109, 14930-14935 (2012)) would be useful.

Response: Thank you very much for the constructive comments. According your suggestion, comparative analyses of BHPF in crystal structures of ER with BPA and BPC were performed. And the following sentences were added or modified in the revised manuscript:

“It was found that BHPF could not be accommodated by the ligand pockets of estrogen receptor α (PDB IDs 1ERE, 3UU7, 3UUA, and 3UUC), but BHPF could be well fitted into the antagonist pocket of estrogen receptor α (PDB ID 3ERT).” (pages 5-6, lines 109-112)

“Interestingly, even though BHPF could not be accommodated by the bisphenol C (BPC) pocket of the estrogen receptor α structure (PDB ID 3UUC), the optimal position of BHPF in estrogen receptor α (PDB ID 3ERT) was very similar to that of BPC in the estrogen receptor α structure (3UUC)²³. It was previously reported that BPC displayed almost full antagonistic activity in the presence of E₂, and that estrogen receptor α with BPC displayed an antagonist conformation similar to that of the OHT-bound structure (3ERT)²³.” (page 13, lines 268-274)

“The three-dimensional structures of the ligand-binding domain of human estrogen receptor α , PDB IDs 1ERE, 3UU7, 3UUA, 3UUC, and 3ERT, were downloaded from

the Protein Data Bank website (<http://www.rcsb.org.pdb>) and used to evaluate the binding affinities of BHPF in the agonist and antagonist pockets of human estrogen receptor α , respectively. ” (page 22, lines 472-476)

REVIEWERS' COMMENTS:

Reviewer #1 (Remarks to the Author):

The authors have responded satisfactorily to this reviewers comments, including additional data, re-working elements of the manuscript, and providing more details about methods.

Reviewer #2 (Remarks to the Author):

The authors have addressed most of my concerns. Just a few points remain.

1. I believe the authors misunderstood my comment about terminology of 5 mL/kg BW BHPF. This dose is meaningless without knowing the concentration of BHPF. Please specify the dosages in mg/kg BW, not mL/kg BW.
2. Provide information on numbers of mice per group for each of the animal experiments. This is provided in some places but not all.
3. The authors have provided background on the uterotrophic assay, but despite its being used by the EPA, most academic researchers in the US think this is a crude assay. I do not ask the authors to do this work over. Instead, in interpreting your results, you need to acknowledge the well-accepted limitations of the uterotrophic and the anti-uterotrophic assay, and do not over-interpret your results. I continue to feel that this assay is crude and of limited utility and should be discussed accordingly.

Reviewer #4 (Remarks to the Author):

I am satisfied with the clarifications and my recommendation is to accept this paper.

Response to Reviewer #2's Comments:

Dear Reviewer #2:

Thank you very much for your comment and suggestion. The comment and suggestion are very helpful for revising and improving our manuscript. We have revised our manuscript according to your comment and suggestion point by point.

Comment 1. I believe the authors misunderstood my comment about terminology of 5 mL/kg BW BHPF. This dose is meaningless without knowing the concentration of BHPF. Please specify the dosages in mg/kg BW, not mL/kg BW.

Response: According to your comment, we checked the terminology of 5 mL/kg BW BHPF and now all the dosages are presented in “mg/kg BW” in the revised manuscript.

Comment 2. Provide information on numbers of mice per group for each of the animal experiments. This is provided in some places but not all.

Response: According to your suggestion, we checked the numbers of mice per group for each of the animal experiment and provided all the numbers in the revised manuscript.

Comment 3. The authors have provided background on the uterotrophic assay, but despite its being used by the EPA, most academic researchers in the US think this is a crude assay. I do not ask the authors to do this work over. Instead, in interpreting your results, you need to acknowledge the well-accepted limitations of the uterotrophic and the anti-uterotrophic assay, and do not over-interpret your results. I continue to feel that this assay is crude and of limited utility and should be discussed accordingly.

Response: The uterotrophic bioassay originated in the 1930's (1-2) and was first standardized for screening by an expert committee in 1962 (3-4). We agree that the uterotrophic and the anti-uterotrophic assays have limitations and sometimes are crude. We think that multiple factors, such as the strain, age, and endogenous estrogen levels of experimental animal, may affect the validity of the assay, a successful utility of the assay requires appropriate experimental conditions, as many other bioassay do, and the animal test system of uterotrophic assay should be validated prior to animal test. According to your suggestion, the following sentences were added in the revised manuscript:

“The uterotrophic assay has been served as an in vivo screening test for estrogenic and anti-estrogenic substances since 1930’s. It is known that multiple factors, such as the strain, age, and endogenous estrogen levels of experimental animal, affect the validity of the assay. So, the animal test system of uterotrophic assay should be validated prior to animal test. In this study, we validated the animal test system of uterotrophic assay using FULV, a full estrogen receptor antagonist, and then evaluated the anti-estrogenicity of BHPF, we found that...”